# OPTIMAL COMMUNITY DETECTION WITH GRAPHICAL NEURAL NETWORKS

## ABSTRACT

This paper investigates the theoretical optimality of community detection in networks using graph neural networks (GNNs). We show that appropriately designed GNNs for supervised community detection can match the performance of classical spectral and likelihood-based methods, achieving information-theoretic optimality under the stochastic block model (SBM). These results provide the first rigorous connection between deep learning algorithms and their statistical guarantees for community detection. We extend existing GNN-based methods into a two-stage framework, where the second stage is critical for ensuring theoretical optimality. Our algorithm is trained on synthetic and/or real-world graphs with known community labels and can be subsequently applied as generic algorithms to any network in an off-the-shelf manner, offering strong practicality. Extensive experiments on both synthetic and real-world datasets support our theoretical findings, demonstrating that the proposed two-stage GNN framework delivers high accuracy and remains robust under model mis-specification. These results establish GNNs as both a theoretically sound and practically effective approach to community detection.

## 1 INTRODUCTION

Community detection is a central task in network analysis, with broad implications across disciplines such as sociology, biology, computer science, and physics. Advances in community detection contribute not only to theoretical developments in graph theory and machine learning but also to practical applications across scientific, industrial, and societal contexts. Over the past decade, research on community detection has seen rapid and substantial progress. Under canonical models such as the stochastic blockmodel (SBM) and degree-corrected blockmodel (DCBM), existing work have investigated thoroughly e.g., recoverability conditions, information-theoretic thresholds and minimax misclassification rates (Abbe et al., 2016; Zhang & Zhou, 2016; Yan, 2016; Gao & Ma, 2021; Gao et al., 2018; Mossel et al., 2023).

When considered in the context of community detection, algorithmic advances outpace theoretical developments in deep learning. On the algorithmic front, there is a growing use of deep learning algorithms in community detection beyond traditional statistical methods. These algorithms include e.g., graph convolutional network (GCN) (Kipf & Welling, 2017; Wang et al., 2021; Liu et al., 2023), graph neural network (GNN) (Chen et al., 2019; Sun et al., 2021; Jiang & Ke, 2023), graph autoencoders (Kipf & Welling, 2016; He et al., 2022). To further improve the effectiveness of representation learning, attention mechanisms have also been incorporated into the neural networks (Veličković et al., 2018; Wang et al., 2023; Zhao et al., 2022). We refer interested readers to the survey paper Su et al. (2024). While these deep learning algorithms have significantly advanced the effectiveness of community detection, not enough attention has been devoted to developing theoretical understandings of their performance. Chen et al. (2019) state the analogy of GNN with the power iteration method, but rigorous analysis on approximation error is absent. Their work also does not answer the question of whether GNN can attain good theoretical bounds in terms of misclassification rate. To the best of the authors' knowledge, there has not been any solid or comprehensive theoretical analysis on the statistical properties of deep learning methods for community detection.

In the broader field of learning, a body of work has established results concerning the statistical properties that deep neural networks can achieve. Yarotsky (2017) is one of the pioneering works

to study the approximation bounds of deep ReLU networks. Schmidt-Hieber (2020) illustrate how ReLU networks' depth and sparsity govern approximation power and convergence rates in non-parametric regression setting. Under such setting, Hu et al. (2021) and Suh et al. (2022) study the generalization properties in overparametrized ReLU neural networks. There are also several results on convergence rates of deep neural networks in classification setting (Kim et al., 2021; Bos & Schmidt-Hieber, 2022; Meyer, 2023), and density estimation setting (Bos & Schmidt-Hieber, 2024). Readers are referred to the survey paper Suh & Cheng (2025) for more details. All these studies focus on deep neural networks with conventional problems such as regression and classification, rather than graph-structured data considered in this paper. Nonetheless, they constitute one of the major motivations for the present work.

This paper aims to close the gap between deep learning-based community detection algorithms and their theoretical properties. We try to answer the following theoretical questions in the community detection context: (i) Can a well-designed GNN perform approximately the computations that are needed in traditional statistical methods, and if so, what is the network depth requirement to reach certain level of approximation accuracy? (ii) Can GNN achieve the minimax rate of community detection in classical models such as SBM? (iii) How good is the trained GNN on the unseen samples? In other words, can the trained GNN achieve strong performance for out-of-sample networks? By addressing these questions, we establish, possibly for the first time in the statistical community, the theoretical properties of deep learning–based methods for community detection. Furthermore, this paper improves upon the existing GNN community detection algorithm by leveraging insights from established statistical theory. In particular, we incorporate a second stage GNN devised to carry out local refinement of normalized edge counting that improves the accuracy of community assignments. This stage is essential for the GNN to achieve the minimax rate.

There are several technical challenges in our theoretical analysis. First, while existing frameworks for conventional deep neural networks offer a comprehensive set of theoretical tools, GNN exhibits substantial differences from those conventional neural networks designed for regression-type data. In particular, GNN computations involve more intricate operations, such as spectral decomposition (or more specifically, orthogonal iteration) on matrices, whose approximation errors and convergence properties that has not been established in prior literature. Second, by establishing a generalization bound, we go beyond the typical existing analysis focused on the error rate only on training (in-sample) data, and prove rigorously that the trained GNN performs well also on out-of-sample networks.

**Contributions.** We summarize the main contributions of this paper as follows:

- We establish, for the first time in the literature, a statistical theoretic foundation for deep learning–based community detection algorithms. We derive error bounds for GNN approximations. We demonstrate that GNN, with ReLU-based activations, can achieve the minimax rate. In SBM with a typical parameter setting, the number of layers needed to achieve minimax rate is at most $O((\log n)^c)$, where $n$ is the network size and $c$ is an positive constant. This result bridges the gap between deep learning algorithms and their underlying theoretical guarantees for community detection.

- We propose a two-stage GNN training scheme, where the second stage augments the existing GNN-based supervised community detection with a local refinement stage. This two-stage approach not only enhances empirical performance but, more importantly, guarantees that the resulting estimator attains the statistically minimax rate.

- We provide a reusable framework to establish generalization bounds of the GNN-based community detection algorithms by investigating the complexity of the underlying GNN-based function class.

**Notations.** We write $I_n$ as the identity matrix of size $n$ (or $I$ as the identity matrix in general) and $J_n$ as the $n \times n$ matrix of ones, i.e., $J_n = \mathbf{1}_{n \times n}$. We use $\mathbf{1}_n$ to represent the $n$-dimensional vector of all ones. For a vector $x \in \mathbb{R}^k$, we use $\|x\|$ to denote the Euclidean norm of $x$, and $\|x\|_{\max} = \max_{1 \le j \le k} |x_j|$ to denote its infinity norm. For a matrix $Y \in \mathbb{R}^{n \times k}$, let $Y_{i\cdot}$ and $Y_{\cdot j}$ represent its $i$th row and $j$th column respectively. Also, let $\|Y\|_{\max} = \max_{i,j} |Y_{ij}|$, $\|Y\|_F$ and $\|Y\|_2$ denote its infinity norm, Frobenius norm and spectral norm respectively. Let $\sigma_{\min}(Y)$ be the

smallest singular value of $Y$, and $\mathrm{col}(Y)$ be the column space of $Y$. For a positive integer $n$, we use $[n]$ to denote the set $\{1, \ldots, n\}$. For a set $\mathbb{S}$, we use $|\mathbb{S}|$ to denote its cardinality.

# 2 MODEL AND ALGORITHM

## 2.1 THE SBM SETUP

We consider the classical SBM setup. Assume the undirected network has $n$ nodes and a known number of communities $K$, with $K = O(1)$. The cases of unknown $K$, and/or $K$ with a larger order, are left to future research. Suppose the true community labels are $\sigma = (\sigma_1, \ldots, \sigma_n) \in [K]^n$. The size of the $k$th community is denoted by $n_k = \sum_{i=1}^n \mathbb{1}_{\{\sigma_i = k\}}$. Let $\mathbf{n} = [n_1, \ldots, n_K]^\top$, $n_{\min} = \min_{k \in [K]} n_k$, and $n_{\max} = \max_{k \in [K]} n_k$. The adjacency matrix $A \in \mathbb{R}^{n \times n}$ has $(i, j)$ entry associated with the edge between every pair of nodes such that $A_{ii} = 0$ and

$$A_{ij} = A_{ji} \stackrel{\text{ind.}}{\sim} \mathrm{Bernoulli}(p_{ij}) \quad \text{for } 1 \le i < j \le n, \tag{1}$$

where the underlying probability matrix $P \in \mathbb{R}^{n \times n}$ is defined by

$$P_{ij} = \mathbb{1}_{\{\sigma_i = \sigma_j\}} p + \mathbb{1}_{\{\sigma_i \ne \sigma_j\}} q \tag{2}$$

with $0 < q < p < 1$. The relationship $p > q$ assures that the network is assortative. It is possible to relax (2) to the form $P_{ij} = p_{\sigma_i \sigma_j}$, meaning that the connection probabilities within and between communities may depend on the specific pair of communities involved. However, for the theoretical derivation presented in this paper, we retain the simpler form given in (2). We write the model determined by (1) and (2) as $\mathrm{SBM}(\mathbf{n}, p, q)$.

For estimated labels $\hat{\sigma} = (\hat{\sigma}_1, \ldots, \hat{\sigma}_n)$, we focus on the misclassification rate $\ell_0(\sigma, \hat{\sigma}) = \min_{\pi \in \mathcal{S}_K} \frac{1}{n} \sum_{i \in [n]} \mathbb{1}_{\{\pi(\sigma_i) \ne \hat{\sigma}_i\}}$, where $\mathcal{S}_K$ represents the set of all possible permutations of $[n]$.

Some notations are in order. Assume $A$ has eigenvalues $\lambda_1, \ldots, \lambda_n$ satisfying $|\lambda_1| \ge \cdots \ge |\lambda_n|$, and associated eigenvectors $v_1, \ldots, v_n \in \mathbb{R}^n$. Let $\eta = |\lambda_K|/|\lambda_{K+1}|$. Define $V = [v_1, \ldots, v_n]$, $V_1 = [v_1, \ldots, v_K]$, $V_2 = [v_{K+1}, \ldots, v_n]$, $\Lambda = \mathrm{diag}(\lambda_1, \ldots, \lambda_n)$, $\Lambda_1 = \mathrm{diag}(\lambda_1, \ldots, \lambda_K)$, $\Lambda_2 = \mathrm{diag}(\lambda_{K+1}, \ldots, \lambda_n)$.

## 2.2 THE GNN FRAMEWORK

To conduct supervised community detection, we adopt the line GNN framework in Chen et al. (2019). Suppose the $m$th layer of the GNN has node features of dimension $d_m$, and these node features are presented by a vector $x^{(m)} \in \mathbb{R}^{n \times d_m}$. That is, the $i$th row of $x^{(m)}$ is the features of node $i$. The GNN is characterized by a group of linear operators on $x^{(m)}$, where these linear operators are precisely the multiplication on the left by $n \times n$ matrices. Following the usual notation, we write a graph $G = (V, E)$. For a graph $G$ with size $|V| = n$ and adjacency matrix $A$, we choose the family of $n \times n$ matrices $\mathcal{F}(A) = \{I_n, J_n, D, A, A_1 \ldots, A_h\}$ with some positive integer $h$, in which $D$ is the degree matrix that is diagonal and whose $(i, i)$ entry $D_{ii}$ is the degree of node $i$, and $A_h = \min(1, A^{2^h})$. We only allow $|\mathcal{F}(A)| = O(1)$. Unlike Chen et al. (2019), we have included an additional matrix $J_n$ in $\mathcal{F}(A)$ in our model. This facilitates our theoretical analysis, while leaving the practical results nearly unaffected.

The GNN maps features of one layer to those of the next via linear operators induced by the matrices in $\mathcal{F}(A)$, followed by the ReLU activation function. In particular, it first computes

$$\bar{z}^{(m+1)} = \sum_{O_i \in \mathcal{F}(A)} O_i x^{(m)} \theta_i^{(m)}, \quad z^{(m+1)} = \rho \left( \sum_{O_i \in \mathcal{F}(A)} O_i x^{(m)} \theta_i^{(m)} \right), \tag{3}$$

where $\theta_i^{(m)} \in \mathbb{R}^{d_m \times \frac{d_{m+1}}{2}}$ are GNN parameters, and $\rho(\cdot)$ is the celebrated ReLU function $\rho(z) = \max(0, z)$ with entry-wise action on matrices. Then it concatenates $z^{(m+1)}$ and $\bar{z}^{(m+1)}$ to get the features of the next layer

$$x^{(m+1)} = \left[ z^{(m+1)}, \bar{z}^{(m+1)} \right] \in \mathbb{R}^{n \times d_{m+1}}. \tag{4}$$

We concatenate $\{\theta_i^{(m)} : O_i \in \mathcal{F}(A)\}$ into a vector $\theta^{(m)}$ with length $d_m(d_{m+1}/2)|\mathcal{F}(A)|$, and denote $\theta = \{\theta^{(0)}, \ldots, \theta^{(M)}\}$ as the collection of parameters in each layer. Essentially, a GNN is fully determined by the parameters $\theta$, and does not depend on the particular graph $G$. Define $\|\boldsymbol{d}\|_{\max} = \max_{1 \leq m \leq M} d_m$ the width of the GNN. With chosen initial features $x^{(0)} \in \mathbb{R}^{n \times d_0}$, we write the effect of GNN with parameters $\theta$ on graph $G$ as a mapping $f_{\theta, x^{(0)}}(\cdot)$

$$f_{\theta, x^{(0)}}(G) = \sigma_{\theta^{(M)}} \circ \sigma_{\theta^{(M-1)}} \circ \cdots \circ \sigma_{\theta^{(0)}}(x^{(0)}), \tag{5}$$

where $\sigma_{\theta^{(m)}}$ means the GNN operators defined by (3) and (4) with parameter $\theta^{(m)}$. Note that $x^{(0)}$ does not depend on $\theta$ in any way, but is allowed to be random.

We attach a softmax layer at the end of the GNN to formulate the community assignment. The result of softmax function on each row of $x^{(M)}$, is a probability matrix $\Psi(A, x^{(0)}; \theta) \in \mathbb{R}^{n \times K}$, whose rows all sum up to 1. With this probability matrix, one can determine the estimated community labels $\sigma(A, x^{(0)}; \theta)$ by $\sigma_i(A, x^{(0)}; \theta) = \max_{k \in [K]} \Psi_{i,k}(A, x^{(0)}; \theta)$.

Define the loss function with respect to $G$ as the cross-entropy

$$\ell_1(\sigma, \Psi(A, x^{(0)}; \theta)) = - \min_{\pi \in \mathcal{S}_K} \frac{1}{n} \sum_{i \in [n]} \log \left( \Psi_{i, \pi(\sigma_i)}(A, x^{(0)}; \theta) \right). \tag{6}$$

The sum in (6) can also be regarded as the log-likelihood function of a multinomial logistic regression with $x^{(M)}$ as the design matrix and $\pi(\sigma)$ as the response. When the training set consists of graphs $G_1, \ldots, G_m$ with adjacency matrices $A^{(1)}, \ldots, A^{(m)}$, and initial features $x^{(01)}, \ldots, x^{(0m)}$, the objective of training is to minimize the empirical risk $\hat{R}_m \left( \{A^{(i)}\}_{i=1}^m, \{x^{(0i)}\}_{i=1}^m; \theta \right) = \sum_{i=1}^m \ell_1(\sigma, \Psi(A^{(m)}, x^{(0)}; \theta))/m$.

### 2.3 A TWO-STAGE GNN SCHEME

The classical two-stage algorithm (Gao et al., 2017; 2018; Gao & Ma, 2021; Gao et al., 2022), introduced in the unsupervised community detection context, consists of a spectral clustering stage and a local refinement stage. We summarize its supervised counterpart as Algorithm 2 in Appendix A. The local refinement procedure, as described by lines 4–6 in Algorithm 2, updates community labels according to the community with which each node has the highest proportion of connections. It is repeated $t$ times to ensure sufficient improvement.

In this paper, we introduce a two-stage GNN training scheme based on the GNN framework described in Section 2.2. This training scheme, devised to mimic Algorithm 2, is described as Algorithm 1. The first stage trains a regular GNN. For each graph in the training set, the second stage GNN takes $Z(\tilde{\sigma}) \in \mathbb{R}^{n \times K}$, the one-hot matrix of the estimated labels $\tilde{\sigma}$ from the first stage, as initial features and train another GNN.

---

1: Train the first GNN with initial features $x^{(0)} \in \mathbb{R}^{n \times d_0}$. For each graph $G$ with adjacency matrix $A$ in the training set, let $\tilde{\sigma} = \sigma(A, x^{(0)}; \tilde{\theta})$ be its estimated community labels from the first GNN.

2: For each graph $G$, compute $Z(\tilde{\sigma}) \in \mathbb{R}^{n \times K}$, where $Z_{i,k}(\tilde{\sigma}) = \mathbb{1}_{\{\tilde{\sigma}_i = k\}}$ for $i \in [n]$ and $k \in [K]$.

3: Train the second GNN initial features $Z(\tilde{\sigma})$ for graph $G$.

**Algorithm 1:** A two-stage GNN training scheme for supervised community detection.

---

The testing is also divided into two stages. For a testing graph $A^{\text{test}}$, we use the first trained GNN to obtain its initial label prediction $\tilde{\sigma}^{\text{test}}$. Based on $\tilde{\sigma}^{\text{test}}$, we compute matrix $Z(\tilde{\sigma}^{\text{test}})$. We then apply the second trained GNN, with $Z(\tilde{\sigma}^{\text{test}})$ as initial features, to obtain a renewed label prediction $\hat{\sigma}^{\text{test}}$. When necessary, the second GNN can be applied iteratively, using $Z(\hat{\sigma}^{\text{test}})$ from the last iteration as the input, to obtain the next label prediction $\hat{\sigma}^{\text{test}(2)}$. This process can be repeated several times until a pre-specified number of repetitions is reached or the label prediction is stable.

# 3 THEORETICAL RESULTS

Rohe et al. (2011), Sussman et al. (2012), Lei & Rinaldo (2015) and numerous other studies analyze the consistency of spectral clustering algorithm for community detection. Gao et al. (2017) and Gao & Ma (2021) show that the classical two-stage algorithm achieves the minimax misclassification rate in SBM. Our objective is to show that the two-stage GNN can attain nearly the same misclassification rate in-sample. More importantly, we also establish generalization guarantees to ensure optimal misclassification rate for out-of-sample data, assuming the out-of-sample data follows the same generating mechanism of the training network data.

The following two assumptions are assumed to hold throughout the entire theoretical derivation, so we do not restate the conditions in the theoretical results.

**Assumption 1.** $n_{\min} \geq n/(\beta K)$ and $n_{\max} \leq \beta n/K$, where $\beta$ is an absolute constant.

**Assumption 2.** $n(p-q) \gg \sqrt{\log n}$ and $n(p-q) \gg \sqrt{np}$.

Assumption 1 assures that all $K$ communities are of the same order. Assumption 2 is a condition that assures certain level of assortativity. In the typical setting of $p = a \log n/n$, $q = b \log n/n$ where $a, b$ are absolute constants, it is satisfied. We denote $\Omega_n$ as the parameter space for $\mathrm{SBM}(\mathbf{n}, p, q)$ that satisfy Assumptions 1 and 2.

## 3.1 ERROR BOUND OF GNN APPROXIMATION TO ORTHOGONAL ITERATION

Observe that the multinomial regression in line 2 of Algorithm 2 can take any matrix spanning $\mathrm{col}(V_1)$ as the design matrix, rather than requiring the precise matrix $V_1$. The *orthogonal iteration* method (Golub & Van Loan, 2013), detailed as Algorithm 3 in Appendix A, can be used to construct a matrix with column space close enough to $\mathrm{col}(V_1)$.

Using a heuristic argument, Chen et al. (2019) point out the analogy of GNN with the power iteration method to obtain $v_1, \ldots, v_K$ (i.e. the columns of $V_1$) sequentially. We, on the other hand, rigorously establish that a properly designed GNN can approximate the output of orthogonal iteration with high accuracy. Notably, orthogonal iteration requires less conditions on $A$ and a shallower GNN compared to power iteration.

For a matrix $Q \in \mathbb{R}^{n \times K}$ with orthonormal columns, the distance between $\mathrm{col}(Q)$ and $\mathrm{col}(V_j)$ is measured by $\mathrm{dist}(\mathrm{col}(Q), \mathrm{col}(V_j)) := \|H_Q - H_{V_j}\|_2$ for $j = 1, 2$, where $H_Q$ and $H_{V_j}$ are projection matrices of $Q$ and $V_j$ respectively. We also have $\mathrm{dist}(\mathrm{col}(Q), \mathrm{col}(V_j)) = \|V_{3-j}^\top Q\|_2$, since $V = [V_1, V_2]$ is orthonormal.

Assume one chooses $Q_0$ as initial features of the GNN. We impose the following condition on $Q_0$:

$$\sigma_{\min}\left(\Lambda_1 V_1^\top Q_0\right) \geq n^{-(r-1)} \tag{7}$$

for some $r > 1$. Condition (7) means that $\mathrm{col}(Q_0)$ cannot be too close to $\mathrm{col}(V_2)$, thus must retain certain directions in $\mathrm{col}(V_1)$. This is a sensible assumption, because otherwise it becomes difficult for the orthogonal iteration to generate directions in $\mathrm{col}(V_1)$. In Appendix D, we show that a matrix sampled from the Haar distribution satisfies (7) with high probability.

The following results characterize the error bound of a properly structured GNN in approximating the orthogonal iteration.

**Theorem 1.** *For any $s > 0$ and any $c_0 > 0$, there exists a GNN with parameters $\theta$, such that for any graph $G \sim \mathrm{SBM}(\mathbf{n}, p, q)$, if initial features $Q_0$ satisfies (7) for its adjacency matrix $A$, the GNN produces features of its last layer $\widehat{Q} \in \mathbb{R}^{n \times K}$ that satisfies*

$$\mathrm{dist}(\mathrm{col}(\widehat{Q}), \mathrm{col}(V_1)) \leq n^{-s} \tag{8}$$

*with probability at least $1 - n^{-c_0}$. The depth $M$ for such GNN satisfies*

$$M \leq 8K^2(s+r)^2 r \frac{(\log n)^3}{\xi^2} + 8K^2((K+1)r+s)(s+r)\frac{(\log n)^2}{\xi}, \tag{9}$$

*where $\xi = \log\left(c_2 n(p-q)/\max\{\sqrt{np}, \sqrt{\log n}\}\right)$ with an absolute constant $c_2 > 0$ depending on $c_0$.*

Theorem 1 provides a general error bound for spectral decomposition using GNNs. It also has independent theoretical significance beyond the context of community detection.

**Remark 1.** *The order of $M$ depends on how much larger $n(p-q)$ is than $\max\{\sqrt{np}, \sqrt{\log n}\}$. In the typical setting of $p = a\log n/n$, $q = b\log n/n$ where $a, b$ are absolute constants, we have $\xi \asymp \log\log n$, so that $M$ is upper bounded by $(\log n)^c$ with $c < 3$.*

**Remark 2.** *The convergence of orthogonal iteration relies on the condition that $\eta$ is bounded away from 0. This is guaranteed by the fact the fact that $\|A - P\|_2 \leq c_1\sqrt{np + \log n}$, which holds with probability at least $1 - n^{-c_0}$ (Lei & Rinaldo, 2015). In contrast, as power iteration generates $v_1, \ldots, v_K$ in a sequential manner, it requires $|\lambda_k|/|\lambda_{k+1}|$ to be bounded away from zero for all $k \in [K]$. Hence orthogonal iteration imposes weaker conditions on $A$. Furthermore, the sequential nature of power iteration leads to more severe accumulation of GNN approximation errors. As a result, orthogonal iteration can achieve sufficient accuracy with a shallower GNN.*

## 3.2 THE IN-SAMPLE MISCLASSIFICATION RATES

The GNN introduced in Theorem 1 can be extended by adding one more layer and a softmax output layer to approximate the multinomial regression. This is precisely what the first stage of Algorithm 1 is designed to address. Define $R = [(np + \log n)(p + (K-1)q)^2]/[n^2(p-q)^4]$. We provide theoretical upper bounds on the misclassification rate this extended GNN in Theorem 2.

**Theorem 2.** *For any $c_0 > 0$, there exists a GNN with parameters $\theta'$ and depth $M'$ satisfying (9), such that for any graph $G \sim \mathrm{SBM}(\mathbf{n}, p, q)$ with true labels $\sigma$, by feeding to this GNN initial features $Q_0$ satisfying (7) for $G$'s adjacency matrix $A$, it outputs estimated labels $\sigma(A, Q_0; \theta')$ that satisfy*

$$\ell_0(\sigma, \sigma(A, Q_0; \theta')) \leq c_1' R.$$

*with probability at least $1 - n^{-c_0}$, for some absolute constant $c_1'$ that depends on $c_0$.*

**Remark 3.** *If $p + q \asymp p - q$, then the bound in Theorem 2 becomes $O((np + \log n)/(n^2(p-q)^2))$. Because of Assumption 2, the bound is $o(1)$, which implies consistency of the GNN classification.*

The purpose of the second stage of Algorithm 1 is to devise an emulation to local refinement procedure. Define $I(p, q) = -2\log\left(\sqrt{pq} + \sqrt{(1-p)(1-q)}\right)$, and make the following assumption:

**Assumption 3.** $nI(p, q) \to \infty$.

**Theorem 3.** *Suppose Assumption 3 holds. For any $\epsilon > 0$, there exists a GNN with depth $M''$ and parameters $\theta''$, such that for any graph $G \sim \mathrm{SBM}(\mathbf{n}, p, q)$ with true labels $\sigma$, as long as its initial label estimate $\sigma^{(0)}$ satisfies $\ell_0(\sigma, \sigma^{(0)}) = o(1)$, it holds that*

$$\sup_{(\mathbf{n}, p, q) \in \Omega_n} \mathbb{P}_{\mathbf{n}, p, q}\left(\ell_0(\sigma, \sigma(A, Z(\sigma^{(0)}); \theta'')) \geq \exp\left[-(1-\epsilon)\tilde{n}I(p, q)\right]\right) \to 0,$$

*where $\tilde{n} = n/2$ when $K = 2$ and $\tilde{n} = n/(\beta K)$ when $K \geq 3$. The GNN depth $M''$ satisfy*

$$M'' \leq \frac{3K}{\log 2}\left(-\log\epsilon - \log I(p, q) + \sqrt{p(1-q)}/\sqrt{q(1-p)} + \log 88\right) + 20K.$$

The error rate in Theorem 3 matches the minimax rate derived in Zhang & Zhou (2016), Gao et al. (2017) and Gao & Ma (2021).

**Remark 4.** *In the typical setting of $p = a\log n/n$, $q = b\log n/n$ where $a, b$ are absolute constants, one can show that $I(p, q) \asymp (p-q)^2/p$. If one takes $\epsilon \geq n^{-\delta}$ for $\delta > 0$, then $M''$ is upper bounded by $O(\log n)$.*

The two GNNs constructed in Theorems 2 and 3 can serve as a device for GNNs in approximately executing the classical two-stage algorithm. However, as the training procedure optimizes GNN parameters by minimizing the cross-entropy loss, the parameters of the trained GNNs may differ from those of the constructed GNNs. In general, there is no guarantee that a small misclassification rate leads to a small cross-entropy, as the probabilities may lack enough margin between correct and incorrect label assignments. But in our model, it is possible to obtain a sufficient margin with high probability. We analyze a "truncated version" of cross-entropy, that can be related to misclassification rate. By bounding this truncated cross-entropy, we can derive an upper bound for the full

cross-entropy. When combined with the inequality misclassification rate $\leq$ cross-entropy$/\log 2$, this bound allows us to establish theoretical guarantees on the in-sample misclassification performance.

Assume graphs $G_1, \ldots, G_m$ in the training set are generated i.i.d. following some prior $\pi_G$. Let their adjacency matrices be $A^{(1)}, \ldots, A^{(m)}$, and initial features be $Q_0^{(1)}, \ldots, Q_0^{(m)}$, which satisfy (7) for each graph.

**Theorem 4.** *Assume the first trained GNN in Algorithm 1 has parameters $\tilde{\theta}$, and satisfies $\hat{R}_m\left(\{A^{(i)}\}_{i=1}^m, \{Q_0^{(i)}\}_{i=1}^m; \tilde{\theta}\right) \leq \hat{R}_m\left(\{A^{(i)}\}_{i=1}^m, \{Q_0^{(i)}\}_{i=1}^m; \theta'\right)$, where $\theta'$ are parameters of the GNN constructed in Theorem 2. For any $c_0 > 0$ and any $\epsilon > 0$, if the training sample size $m \geq R^{-(1+\epsilon)}(\log n)^{1+\epsilon}$, the the first trained GNN outputs in-sample estimated community labels $\sigma(A^{(i)}, Q_0^{(i)}; \tilde{\theta})$ that satisfy*

$$\frac{1}{m} \sum_{i=1}^m \ell_0(\sigma, \sigma(A^{(i)}, Q_0^{(i)}; \tilde{\theta})) \leq \tilde{c}_1 R^{1-\epsilon}$$

*with probability at least $1 - n^{-c_0}$, for some constant $\tilde{c}_1$ that depends on $c_0$.*

Let $\tilde{\sigma}^{(i)} = \sigma(A^{(i)}, Q_0^{(i)}; \tilde{\theta})$ be the estimated labels from the first trained GNN for graph $G_i$ in the training set.

**Theorem 5.** *Suppose Assumption 3 holds. Assume the second trained GNN in Algorithm 1 has parameters $\hat{\theta}$, and satisfies $\hat{R}_m\left(\{A^{(i)}\}_{i=1}^m, \{Z(\tilde{\sigma}^{(i)})\}_{i=1}^m; \hat{\theta}\right) \leq \hat{R}_m\left(\{A^{(i)}\}_{i=1}^m, \{Z(\tilde{\sigma}^{(i)})\}_{i=1}^m; \theta''\right)$, where $\theta''$ are parameters of the GNN constructed in Theorem 3. For any $\epsilon > 0$ and any $c_0 > 0$, if the training sample size $m \geq \exp\{2\tilde{n}I(p,q)\}(\log n)^{1+\epsilon}$, then the second trained GNN outputs in-sample estimated community labels $\sigma(A^{(i)}, Z(\tilde{\sigma}^{(i)}); \hat{\theta})$ that satisfy*

$$\frac{1}{m} \sum_{i=1}^m \ell_0(\sigma, \sigma(A^{(i)}, Z(\tilde{\sigma}^{(i)}); \hat{\theta})) \leq \exp\left[-(1-3\epsilon)\tilde{n}I(p,q)\right]$$

*with probability at least $1 - n^{-c_0}$.*

**Remark 5.** *Theorems 4 and 5 hinge on the assumption that the GNN training can effectively decrease the empirical risk. In practice, the convergence of the training is influenced by the optimization landscape (Chen et al., 2019), a topic beyond the scope of the present study.*

### 3.3 GENERALIZATION BOUNDS

We focus on the class of GNN functions.

$$\mathcal{G}(M, \boldsymbol{d}, \mathfrak{s}) := \left\{ f_{\theta, x^{(0)}} \text{ of the form (5)} : \max_{0 \leq m \leq M} \|\theta^{(m)}\|_{\max} \leq 1, \sum_{m=0}^M \sum_{i=1}^{|\mathcal{F}|} \|\theta_k^{(m)}\|_0 \leq \mathfrak{s} \right\}, \quad (10)$$

where we abuse the notation slightly by defining $\|F\|_0 = \sum_{i,j} \mathbb{1}_{\{F_{ij} \neq 0\}}$ for matrix $F$. We call the class of GNN clustering algorithms $\mathcal{SG}(M, \boldsymbol{d}, \mathfrak{s}) := \{\text{softmax} \circ f : f \in \mathcal{G}(M, \boldsymbol{d}, \mathfrak{s})\}$.

The specification of $(M, \boldsymbol{d}, \mathfrak{s})$ follows what the GNN that approximate the orthogonal iterations, which is used to restrict our parameter search in training GNN's.

**Theorem 6.** *Fix $M = O(\log^2(n))$, $\boldsymbol{d}$ with $\|\boldsymbol{d}\|_{\max} = O(n)$ and $\mathfrak{s} = O(n \log(n))$. Under the condition of Theorem 4, by taking $m = O\left(R^{-(1+\varepsilon)} \max((\log^{1+\varepsilon} n), n \log^4(n))\right)$, we have with probability $1 - n^{-c}$ for some $c < 1$, the expected misclassification rate on $A \sim \text{SBM}(\boldsymbol{n}, p, q)$ of the trained GNN characterized by $\tilde{\theta}$ on $\mathcal{SG}(M, \boldsymbol{d}, \mathfrak{s})$ can be bounded*

$$\mathbb{E}[\ell_0(\sigma, \sigma(A, Q; \tilde{\theta}) \mid \tilde{\theta}] \leq c' R^{1-\varepsilon},$$

*where the constant $c'$ depends on $\varepsilon$ and $c$.*

The theorem establishes that the obtained GNN community detection algorithm trained on SBM synthetic data attains the same mischassification rate as in Theorem 4, if the algorithm is applied on SBM networks generated following the same SBM laws. The GNN community detection algorithm, thus, is effective not just on in-sample networks, but also on out-of-sample networks.

## 4 NUMERICAL STUDIES

We conduct comprehensive experiments to assess our two-stage GNN scheme. The objectives are twofold: (i) to benchmark its performance against the baseline GNN model, and (ii) to validate its improved generalization capability when trained on diverse graph models.

### 4.1 SYNTHETIC EXPERIMENTS ON SBM

We adopt the typical setting $p = a \log n/n$, $q = b \log n/n$. Note that $a, b$ are uniquely determined by $C = a + (K-1)b$ and $\text{SNR} = (a-b)^2/[K(a+(K-1)b)]$, where $C$ controls the node degree and SNR represents signal-to-noise ratio. We examine three community counts, $K = 2, 4, 8$, and for each we employ a two-stage training scheme. For a fixed $K$, we construct a training set with 4,500 graphs, by varying the parameters $C$, SNR, and community sizes $\mathbf{n}$. The test set, consisting of 1,800 graphs, is also constructed using combinations of $C$, SNR, and community sizes $\mathbf{n}$. The detailed data generating mechanism and training configuration is described in Appendix I.1.

The performance of the base (one-stage) GNN and two-stage GNN, grouped by SNR and $\mathbf{n}$, are shown in Table 1. The experimental results clearly show that our two-stage GNN method achieves higher accuracy across almost all test scenarios. The two-stage GNN demonstrates particularly pronounced advantages when $K$ is large, the communities are imbalanced, and the SNR is low.

Table 1: Test accuracy of base and two-stage GNN's on the SBM. All values are percentages, reported in the mean (standard deviation) format. $\mathbf{n}^{(1)}, \mathbf{n}^{(2)}, \mathbf{n}^{(3)}, \mathbf{n}^{(4)}$, correspond to balanced, slightly imbalanced, moderately imbalanced, extremely imbalanced community sizes, respectively.

| $\mathbf{n}$ | SNR | $K = 2$ | | $K = 4$ | | $K = 8$ | |
|---|---|---|---|---|---|---|---|
| | | Base | Two-stage | Base | Two-stage | Base | Two-stage |
| | 0.25 | 52.8 (3.26) | 52.9 (3.34) | 47.2 (13.2) | 50.3 (16.1) | 44.1 (1.13) | 50.4 (2.16) |
| | 0.75 | 98.8 (0.06) | 99.0 (0.05) | 98.9 (0.18) | 99.0 (0.08) | 77.1 (0.76) | 82.7 (0.73) |
| $\mathbf{n}^{(1)}$ | 1.50 | 100 (0.00) | 100 (0.02) | 99.9 (0.17) | 100 (0.01) | 81.5 (1.35) | 83.1 (1.33) |
| | 0.25 | 73.1 (8.43) | 75.0 (7.03) | 57.7 (10.7) | 62.8 (11.6) | 47.9 (0.57) | 54.4 (1.55) |
| | 0.75 | 98.9 (0.02) | 99.0 (0.05) | 99.0 (0.12) | 99.0 (0.03) | 79.0 (1.31) | 85.5 (0.55) |
| $\mathbf{n}^{(2)}$ | 1.50 | 100 (0.01) | 100 (0.01) | 100 (0.06) | 100 (0.01) | 81.9 (1.59) | 86.5 (0.66) |
| | 0.25 | 81.4 (4.15) | 83.0 (3.01) | 77.4 (2.77) | 80.4 (1.54) | 68.3 (1.83) | 72.9 (1.71) |
| | 0.75 | 98.9 (0.03) | 99.1 (0.05) | 98.9 (0.10) | 99.0 (0.07) | 87.8 (1.05) | 94.7 (0.66) |
| $\mathbf{n}^{(3)}$ | 1.50 | 100 (0.02) | 100 (0.02) | 100 (0.01) | 100 (0.01) | 90.4 (1.26) | 97.2 (0.36) |
| | 0.25 | 86.3 (2.20) | 87.2 (1.64) | 80.2 (0.40) | 79.7 (0.65) | 51.8 (4.61) | 59.4 (7.98) |
| | 0.75 | 99.1 (0.03) | 99.2 (0.04) | 89.1 (0.82) | 91.2 (0.53) | 63.8 (9.09) | 70.6 (5.81) |
| $\mathbf{n}^{(4)}$ | 1.50 | 100 (0.01) | 100(0.01) | 97.5 (0.38) | 97.3 (0.45) | 80.0 (4.06) | 85.7 (6.31) |

### 4.2 SYNTHETIC EXPERIMENTS ON MIXED MODELS

To assess robustness to model mis-specification, we implement an evaluation protocol using a mixed-data training approach. Specifically, we construct a training set with 4,500 graphs, and a test set with 1,800 graphs, both comprising equally numbered instances from both SBM and DCBM. For both models, we keep the choices of $K$, $C$, SNR and $\mathbf{n}$ the same as the first experiment in Section 4.1. See Appendix I.2 for details.

Table 2 summarizes the performance of this mixed-data training approach, compared with the training only with SBM graphs. Mixed training significantly improves model performance. Compared to training on SBM alone, this mixed approach yields higher accuracy and lower variance across all conditions, indicating a more robust and stable model.

To further evaluate our model's generalization capabilities, we conducted additional experiments by training on the latent space model (LSM). The results demonstrated that the model trained on LSM achieved a performance similar to that of the SBM + DCBM mixed-trained model. For a comprehensive overview of these findings, see Appendix I.2.

Table 2: Test accuracy under different training sets tested on mixed SBM and DCBM graphs. All values are percentages, reported in the mean (standard deviation) format.

| Training set | SNR | $K=2$ | | $K=4$ | | $K=8$ | |
|---|---|---|---|---|---|---|---|
| | | Base | Two-stage | Base | Two-stage | Base | Two-stage |
| SBM | 0.25 | 68.4 (13.0) | 69.4 (13.5) | 51.3 (19.7) | 53.5 (20.5) | 42.6 (18.0) | 46.7 (20.0) |
| | 0.75 | 82.3 (18.7) | 82.1 (19.1) | 74.6 (24.2) | 75.8 (23.8) | 65.6 (15.8) | 72.6 (13.8) |
| | 1.50 | 89.0 (16.0) | 86.6 (17.3) | 90.2 (11.1) | 92.8 (9.1) | 76.1 (9.2) | 81.7 (8.8) |
| SBM + DCBM | 0.25 | 74.6 (11.1) | 75.7 (10.8) | 64.8 (8.2) | 67.9 (7.9) | 44.3 (11.7) | 49.8 (9.3) |
| | 0.75 | 97.0 (2.1) | 97.2 (2.0) | 94.0 (4.6) | 94.7 (3.9) | 72.0 (5.5) | 83.8 (4.5) |
| | 1.50 | 99.2 (0.9) | 99.2 (0.8) | 98.3 (1.6) | 98.5 (1.4) | 79.3 (3.7) | 89.3 (4.7) |

## 4.3 REAL DATA EXPERIMENTS

We evaluate the proposed method on five real-world networks: the political blog network (Adamic & Glance, 2005) with $n = 1,222$, $K = 2$; the Simmons College network with $n = 1,137$, $K = 4$ and the Caltech network with $n = 590$, $K = 8$ (Traud et al., 2011; 2012)), both preprocessed following (Chen et al., 2018); a manufacturing company network (Weng & Feng, 2022) with $n = 74$, $K = 4$; and the French high school friendship network (Mastrandrea et al., 2015) with $n = 329$, $K = 9$.

As in Section 4.2, we introduce the model structures and compare training only based on SBM graphs and training based on SBM+DCBM graphs. Table 3 summarizes the resulting accuracies on these datasets. Models trained on SBM+DCBM consistently outperform those trained solely on SBM. We notice that these real-world networks exhibit certain levels of heterogeneity, a structure not adequately captured by SBM. By incorporating DCBM into the training data, the model learns to take into account degree heterogeneity, which improves its generalization ability to real data. Furthermore, applying the two-stage GNN consistently improves performance on these datasets.

Table 3: Test accuracy under different training schemes tested on real datasets.

| Dataset | SBM | | SBM+DCBM | |
|---|---|---|---|---|
| | Base | Two-stage | Base | Two-stage |
| Political Blog | 89.2% | 93.3% | 94.8% | 95.3% |
| Simmons | 73.0% | 73.7% | 73.3% | 77.5% |
| Caltech | 46.9% | 62.7% | 70.3% | 74.6% |
| Company | 94.5% | 94.5% | 94.5% | 96.0% |
| High school | 73.6% | 85.4% | 89.4% | 98.5% |

## 5 CONCLUSION

In this paper we establish a rigorous theoretical foundation for GNNs in supervised community detection, showing that they can achieve information-theoretic optimality while remaining effective and robust in practice. The proposed two-stage GNN framework not only bridges the gap between deep learning and classical statistical methods but also offers a practical and versatile tool for analyzing real-world networks.

A natural way of extending our model setup is to study DCBM and latent space model (LSM). Prior work has shown that the two-stage algorithm achieves the minimax rates under both DCBM (Gao et al., 2018) and LSM (Gao et al., 2022). This provides a basis for future theoretical development of GNNs on these models. Another interesting direction is to establish theoretical guarantees for unsupervised community detection with deep neural networks. Unsupervised learning requires the network to emulate clustering rather than classification. Developing principled methods and analyses in this context would be a new venue to study GNN-based methods and strengthen their theoretical foundations. We omitted a detailed examination of the training landscape for GNNs. Characterizing the basin of attraction leading to estimators of statistical precision would further enhance our findings.

ACKNOWLEDGMENTS

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

## A    DESCRIPTIONS RELEVANT ALGORITHMS

The supervised version of the classical two-stage algorithm for community detection is described as follows:

1: Perform spectral decomposition on $A$ to get $V_1$.
2: Fit a multinomial regression with $V_1$ as the design matrix and $\sigma$ as the response. Let $\sigma^{(0)}$ be the fitted community labels.
3: **for** $\tau = 1, 2, \ldots t$ **do**
4:    **for** $i = 1, 2, \ldots, n$ **do**
5:       Let $\sigma_i^{(\tau)} = \arg\max_{k \in [K]} \frac{1}{|\{j : \sigma_j^{(\tau-1)} = k\}|} \sum_{\{j : \sigma_j^{(\tau-1)} = k\}} A_{ij}$.
6:    **end for**
7: **end for**
8: Output $\sigma^{(t)} = (\sigma_1^{(t)}, \ldots, \sigma_n^{(t)})$.

**Algorithm 2:** The classical two-stage algorithm for supervised community detection.

The orthogonal iteration method (Golub & Van Loan, 2013) is described as follows:

1: Initialize with $Q_0 \in \mathbb{R}^{n \times K}$, which has orthonormal columns.
2: **for** $t = 1, 2, \ldots$ **do**
3:    Compute $Y_t = AQ_{t-1}$.
4:    Implement QR decomposition $Y_t = Q_t R_t$, where $Q_t \in \mathbb{R}^{n \times K}$ has orthonormal columns, and $R_t \in \mathbb{R}^{K \times K}$ is upper-triangular.
5: **end for**
6: Output $Q_t$.

**Algorithm 3:** Orthogonal iteration

## B    ERROR BOUNDS OF GNN APPROXIMATIONS TO BASIC ARITHMETIC OPERATORS

Lemma 7 provides the basic building blocks to analyze the approximation errors of GNNs. Our analysis leverages techniques from Schmidt-Hieber (2020); Bos & Schmidt-Hieber (2024).

**Lemma 7** (Basic arithmetic operations with GNN). *Suppose we have $v, y \in \mathbb{R}^{n \times 1}$.*

1. *(Inner product) For $\|v\|_{\max}, \|y\|_{\max} \leq \kappa$ for some integer $\kappa$ and some width configuration $\boldsymbol{d}$ such that $\|\boldsymbol{d}\|_{\max} = \max(22, 2\kappa^2 + 6)$, there exists a GNN architecture $\mathcal{G}(m+10, \boldsymbol{d}, 41m+9\kappa^2+179, (v, y, \mathbf{1}))$ that maps $(v, y, \mathbf{1}_{n \times 1})$ to $(\widetilde{\langle v, y \rangle} \mathbf{1}_{n \times 1}, v, y, \mathbf{1}_{n \times 1})$ such that $|\widetilde{\langle v, y \rangle} - \langle v, y \rangle| \leq 4n\kappa^2 2^{-m}$.*

2. *(Column norm) Suppose $\|v\|_{\max} \leq \kappa$ for some integer $\kappa$ and $\|v\|_2 \geq 2\varepsilon$ for some fixed $\varepsilon$. Assume $4n\kappa^2 2^{-m} < \varepsilon < 1$. There exists a GNN architecture $\mathcal{G}(M, \boldsymbol{d}, \mathfrak{s}, (v, v, \mathbf{1}))$ that maps $(v, v, \mathbf{1}_{n \times 1})$ to $(\widetilde{\|v\|} \mathbf{1}_{n \times 1}, v, \mathbf{1}_{n \times 1})$, where $M = 3m+23$, $\|\boldsymbol{d}\|_{\max} \leq \max(22, 2k^2 + 6, 24 \cdot 2^m + 6)$, $\mathfrak{s} \leq 47m + 9\kappa^2 + 11421(2m + 6)2^m + 218$ and $|\widetilde{\|v\|_2} - \|v\|_2| \leq (2n\kappa^2 + 38)\varepsilon^{-1}2^{-m}$.*

3. *(Inversion) Suppose for $u \in \mathbb{R}^{n \times 1}$, $\min_i u_i \geq \varepsilon$ with $2^{-m} \leq \varepsilon \leq 1$. Suppose $u^{-1} = (u_i^{-1})_{i=1}^n$. There exists a GNN architecture $\mathcal{G}(M, \boldsymbol{d}, \mathfrak{s}, (u, v, \mathbf{1}))$ that maps $(u, v, \mathbf{1}_{n \times 1})$ to $(\widetilde{u^{-1}}, v, \mathbf{1}_{n \times 1})$, where $M = 2m + 13$, $\|\boldsymbol{d}\|_{\max} \leq 24 \cdot 2^m + 6$, $\mathfrak{s} \leq 11421(2m + 6)2^m + 6m + 39$ and $\|\widetilde{u^{-1}} - u^{-1}\|_{\max} \leq 57\varepsilon^{-2}2^{-m}$.*

*Proof of Lemma 7.* With loss of generality, we assume $\kappa$ is an integer, since otherwise we can always take a new upper bound $\lceil \kappa \rceil$.

For the family of matrices $\mathcal{F} = \{I_n, \mathbf{1}_{n \times n}, D, \dots\}$, we always denote $O_1 = I$, $O_2 = \mathbf{1}_{n \times n}$ and $O_3 = D$.

We prove the first claim on the inner product. Set the starting state $x^{(0)} = (v, y, \mathbf{1})$. We set $\theta_i^{(0)} = \mathbf{0}$ for all $i \in \{2, 3, \dots\}$ and

$$
\theta_1^{(0)} = \begin{pmatrix}
0 & 0 & 0 & 0 & 0 \\
0 & 0 & 0 & 0 & 0 \\
0 & 0 & 0 & 0 & 0 \\
1/2\kappa & 0 & 0 & 1 & 0 \\
0 & 1/2\kappa & 0 & 0 & 1 \\
1/2 & 1/2 & 1 & 0 & 0
\end{pmatrix},
$$

and correspondingly $\bar{z}^{(1)}$ is $(\tilde{v}, \tilde{y}, \mathbf{1}, v, y) := (v/2\kappa + 1/2, y/2\kappa + 1/2, \mathbf{1}, v, y)$.

For the next layer of the GNN, we set $\theta_i^{(1)} = \mathbf{0}$ for all $i \neq 1$ and

$$
\theta_1^{(1)} = \begin{pmatrix}
\mathbf{0}_{5\times 5} & & \mathbf{0}_{5\times 4} \\
 & & \begin{pmatrix} 0 & 1 & 0 & 0 & 0 & 0 \\ 0 & 0 & 1 & 0 & 0 & 0 \\ 1 & 0 & 0 & 1 & 0 & 0 \\ 0 & 0 & 0 & 0 & 1 & 0 \\ 0 & 0 & 0 & 0 & 0 & 1 \end{pmatrix} \\
 & K_1 & 
\end{pmatrix}
\quad \text{with} \quad
K_1 = \begin{pmatrix}
1/4 & 1/2 & 1/2 & 1/4 & 1/2 \\
-1/4 & -1/2 & 1/2 & 1/2 & 1/2 \\
-1/4 & 0 & 0 & 0 & -1/2 \\
0 & 0 & 0 & 0 & 0 \\
0 & 0 & 0 & 0 & 0
\end{pmatrix}.
$$

The first row of $\bar{z}^{(2)}$ is

$$
\left(\tilde{v}_1/4 - \tilde{y}_1/4 - 1/4, \tilde{v}_1/2 - \tilde{y}_1/2, \tilde{v}_1/2 + \tilde{y}_1/2, \tilde{v}_1/4 + \tilde{y}_1/4, \tilde{v}_1/2 + \tilde{y}_1/2 - 1/2, 1, \tilde{v}_1, \tilde{y}_1, 1, v, y\right)
$$

The first row of $z^{(2)}$ is

$$
\left(T_+\left(\tfrac{\tilde{v}_1 - \tilde{y}_1 + 1}{2}\right), T_-^1\left(\tfrac{\tilde{v}_1 - \tilde{y}_1 + 1}{2}\right), \rho\left(\tfrac{\tilde{v}_1 + \tilde{y}_1}{2}\right), T_+\left(\tfrac{\tilde{v}_1 + \tilde{y}_1}{2}\right), T_-^1\left(\tfrac{\tilde{v}_1 + \tilde{y}_1}{2}\right), 1, \rho(\tilde{v}_1), \rho(\tilde{y}_1), 1, \rho(v_1), \rho(y_1)\right)
$$

where $T^k : [0, 2^{2-2k}] \to [0, 2^{-2k}]$ and $T_-^k$ defined by

$$
T_-^k(x) := \rho(x - 2^{1-2k}),
$$
$$
T_+(x) := \rho(x/2),
$$
$$
T^k(x) := (x/2) \wedge (2^{1-2k} - x/2) = T_+(x) - T_-^k(x).
$$

Combine both $z^{(2)}$ and $\bar{z}^{(2)}$ for $x^{(2)}$.

Furthermore, for $t = 2, \dots, m+4$,

$$
\theta_1^{(t)} = \begin{pmatrix}
K_t & \mathbf{0}_{6\times 5} \\
\mathbf{0}_{5\times 6} & \mathbf{0}_{5\times 5} \\
\mathbf{0}_{6\times 6} & \mathbf{0}_{6\times 5} \\
\mathbf{0}_{5\times 6} & I_5
\end{pmatrix}, \quad \text{and} \quad \theta_i^{(t)} = \mathbf{0}, \forall i \neq 1,
$$

where $K_t \in \mathbb{R}^{6\times 6}$ is the corresponding weight matrix arising from the NN setup in Schmidt-Hieber (2020, Lemma A.2), with the only change of the role of the constant $1/4$ term being replaced by our constant term $1$. For $t = m + 5$,

$$
\theta_1^{(m+5)} = \begin{pmatrix}
K_{m+5} & \mathbf{0}_{6\times 5} \\
\mathbf{0}_{5\times 1} & \mathbf{0}_{5\times 5} \\
\mathbf{0}_{6\times 1} & \mathbf{0}_{6\times 5} \\
\mathbf{0}_{5\times 1} & I_5
\end{pmatrix}, \quad \text{and} \quad \theta_i^{(m+5)} = \mathbf{0}, \forall i \neq 1,
$$

where $K_{m+5} \in \mathbb{R}^{6\times 1}$. Applying Schmidt-Hieber (2020, Lemma A.2), the first row of $(z^{(m+6)})$ arrives at

$$
(\widetilde{\tilde{v}_1 \tilde{y}_1}, \rho(\tilde{v}_1), \rho(\tilde{y}_1), 1, \rho(v_1), \rho(y_1)),
$$

where $|\widetilde{\tilde{v}_1 \tilde{y}_1} - \tilde{v}_1 \tilde{y}_1| \leq 2^{-m}$. In other words, the first column of $z^{(m+6)}$ is the approximate element-wise product of $\tilde{v}$ and $\tilde{y}$. For the first row of $\bar{z}^{(m+7)}$, the last five elements are $(\tilde{v}_1, \tilde{y}_1, 1, v_1, y_1)$, and the first element is the value of $\widetilde{\tilde{v}_1 \tilde{y}_1}$ before being applied the ReLU activation function (and the

value will be discarded immediately in the next GNN layer). We now go back to $v_1 y_1$. Keeping in mind $v_1 y_1 = \kappa^2 (4\tilde{v}_1 \tilde{y}_1 - 2\tilde{v}_1 - 2\tilde{y}_1 + 1)$, we devise the following two layers:

$$\theta_1^{(m+6)} = \begin{pmatrix} \mathbf{1}_{1\times 4} & \mathbf{0}_{1\times 7} \\ \mathbf{0}_{5\times 4} & \mathbf{0}_{5\times 7} \\ & \begin{pmatrix} 0 & 0 & 0 & 0 & 0 & 0 & 0 \\ -1 & -1 & 0 & 0 & 0 & 0 & 0 \\ 0 & 0 & -1 & -1 & 0 & 0 & 0 \\ 0 & 0 & 0 & 0 & 1 & 0 & 0 \\ 0 & 0 & 0 & 0 & 0 & 1 & 0 \\ 0 & 0 & 0 & 0 & 0 & 0 & 1 \end{pmatrix} \end{pmatrix}, \quad \text{and } \theta_i^{(m+6)} = \mathbf{0}, \forall i \neq 1.$$

$$\theta_1^{(m+7)} = \begin{pmatrix} \mathbf{0}_{11\times \kappa^2} & \mathbf{0}_{11\times 3} \\ \mathbf{1}_{8\times \kappa^2} & \mathbf{0}_{8\times 3} \\ \mathbf{0}_{3\times \kappa^2} & I_3 \end{pmatrix}, \quad \text{and } \theta_i^{(m+7)} = \mathbf{0}, \forall i \neq 1.$$

The first row of the resulting state $\bar{z}^{(m+8)}$ is $(4\widetilde{\tilde{v}_1 \tilde{y}_1} - 2\tilde{v}_1 - 2\tilde{y}_1 + 1, \ldots, 4\widetilde{\tilde{v}_1 \tilde{y}_1} - 2\tilde{v}_1 - 2\tilde{y}_1 + 1, 1, v_1, y_1)$, where $4\widetilde{\tilde{v}_1 \tilde{y}_1} - 2\tilde{v}_1 - 2\tilde{y}_1 + 1$ is repeated $\kappa^2$ times. We proceed to arrange another rescaling layers of GNN as follows:

$$\theta_1^{(m+8)} = \begin{pmatrix} \mathbf{0}_{(\kappa^2+3)\times 1} & \mathbf{0}_{(\kappa^2+3)\times 3} \\ \mathbf{1}_{\kappa^2\times 1} & \mathbf{0}_{\kappa^2\times 3} \\ \mathbf{0}_{3\times 1} & \begin{pmatrix} 0 & 0 & 1 \\ 1 & 0 & 0 \\ 0 & 1 & 0 \end{pmatrix} \end{pmatrix}, \quad \text{and } \theta_i^{(m+8)} = \mathbf{0}, \forall i \neq 1.$$

The resulting $\bar{z}^{(m+9)}$ has its first row as $(\widetilde{v_1 y_1}, v_1, y_1, 1)$, where $|\widetilde{v_1 y_1} - v_1 y_1| \leq 4\kappa^2 2^{-m}$.

In the next GNN layer, we get the approximate inner product $\langle v, y \rangle$ by setting $\theta_i^{(m+9)} = \mathbf{0}$ for all $i \notin 1, 2$ and

$$\theta_2^{(m+9)} = \begin{pmatrix} 0 & \mathbf{0}_{4\times 4} \\ 1 & \mathbf{0}_{1\times 4} \\ \mathbf{0}_{3\times 1} & \mathbf{0}_{3\times 4} \end{pmatrix}, \quad \theta_1^{(m+9)} = \begin{pmatrix} \mathbf{0}_{5\times 1} & \mathbf{0}_{5\times 1} & \mathbf{0}_{5\times 3} \\ \mathbf{0}_{3\times 1} & \begin{pmatrix} 1 \\ 0 \\ 0 \end{pmatrix} & I_3 \end{pmatrix}.$$

) We now have $\bar{z}^{(m+10)} = (\widetilde{\langle v, y \rangle} \mathbf{1}_{n\times 1}, v, v, y, \mathbf{1}_{n\times 1})$, where $|\widetilde{\langle v, y \rangle} - \langle v, y \rangle| < 4n\kappa^2 2^{-m}$.

For the second claim, we first apply the first claim, that there exists a GNN architecture $\mathcal{G}(m + 10, \boldsymbol{d}, 41m + 9\kappa^2 + 179, (v, v, \mathbf{1}))$ mapping $(v, v, \mathbf{1})$ to $(\widetilde{\|v\|_2^2}\mathbf{1}, v, v, \mathbf{1})$ such that $|\widetilde{\|v\|_2^2} - \|v\|_2^2| \leq 4n\kappa^2 2^{-m}$, where $\|\boldsymbol{d}\|_{\max} = \max(22, 2\kappa^2 + 6)$. Now we need to take the square root. Note $f : x \in [\varepsilon^2, \infty) \mapsto \sqrt{x} \in [\varepsilon, \infty)$ has at Hölder smoothness 1 with radius $\varepsilon^{-1}/2$. Applying Schmidt-Hieber (2020, Theorem 5) and using $\varepsilon > 4n\kappa^2 2^{-m}$, we can build a GNN of $2m + 13$ layers with maximal width $12 \cdot 2^m + 6$ and sparsity $\mathfrak{s} \leq 11421(2m+6)2^m + 6m + 39$ such that the output is $(\widetilde{\|v\|_2}\mathbf{1}, v, \mathbf{1})$ and $|\widetilde{\|v\|_2} - \sqrt{\widetilde{\|v\|_2^2}}| \leq 38\varepsilon^{-1}2^{-m}$. Combining the two GNNs gives the statement.

We show the third claim. Assumption $2^{-m/2} \leq \varepsilon \leq 1$ and $\min_i u_i \geq \varepsilon$. Note $f : x \in [\varepsilon, \infty) \mapsto x^{-1} \in [\varepsilon^{-1}, 0)$ has Hölder smoothness 1 with radius $\varepsilon^{-2}$. For $(u, v, \mathbf{1}_{n\times 1})$ Apply Schmidt-Hieber (2020, Theorem 5) again, and using we can explicitly construct a GNN of $2m + 13$ layers with maximal width $12 \cdot 2^m + 6$ and sparsity $\mathfrak{s} \leq 11421(2m+6)2^m + 6m + 39$ such that the first row of the output is $(\widetilde{u_1^{-1}}, v_1, 1)$ with $\max_i |\widetilde{u_i^{-1}} - u_i^{-1}| \leq 57\varepsilon^{-2}2^{-m}$. $\square$

## C    PROOF OF THEOREM 1

**Properties of orthogonal iteration.** We first provide a convergence rate result for orthogonal iteration. Assume the orthogonal iteration, described as Algorithm 3, takes $Q_0$ as the initial value. Let $d_t = \|V_2^\top Q_t\|_2 = \text{dist}(\text{col}(Q_t), \text{col}(V_1))$ for $t = 0, 1, \ldots$, which we expect to get small for sufficiently large $t$.

**Lemma 8.** *Assume $Q_0$ satisfies (7). For any $s > 0$, with $T$ defined by*

$$T = \left\lceil \frac{(s+r)\log n + \log 2}{\log \eta} \right\rceil, \tag{11}$$

*Algorithm 3 outputs $Q_T$ that satisfies $d_T \leq n^{-s}/2$.*

*Proof.* The QR decomposition of $Y_t$ is equivalent to the Gram-Schmidt process applied to the columns of $Y_t$. Denote the $k$th column of $Y_t$ by $y_{k,t}$. The Gram-Schmidt process has the following steps:

- The first step:

$$q_{1,t} = \frac{y_{1,t}}{\|y_{1,t}\|}. \tag{12}$$

- The $k$th step ($k = 2, \ldots, K$):

$$u_{k,t} = y_{k,t} - \sum_{j=1}^{k-1} \langle y_{k,t}, q_{j,t} \rangle q_{j,t}, \tag{13}$$

$$q_{k,t} = \frac{u_{k,t}}{\|u_{k,t}\|}. \tag{14}$$

Then, $Q_t = [q_{1,t}, \cdots, q_{K,t}]$ is the Q-component of the QR decomposition of $Y_t$.

By the description of Algorithm 3, we have $Q_t(R_t R_{t-1} \cdots R_1) = A^t Q_0$. Denote $S_t = R_t R_{t-1} \cdots R_1$. Since $A^t = V \Lambda^t V^\top$, then

$$V^\top Q_t S_t = \Lambda^t V^\top Q_0.$$

By the block structure of $V$ and $\Lambda$, we get

$$\Lambda_1^t V_1^\top Q_0 = V_1^\top Q_t S_t, \quad \Lambda_2^t V_2^\top Q_0 = V_2^\top Q_t S_t.$$

Letting $V_j^\top Q_t = W_{j,t}$ for $j = 1, 2$ and $t = 0, 1, \ldots$, it then follows that

$$W_{2,t} = \Lambda_2^t W_{2,0} W_{1,0}^{-1} \Lambda_1^{-t} W_{1,t}. \tag{15}$$

Since $\sigma_{\min}(\Lambda_1 W_{1,0}) \leq |\lambda_1| \sigma_{\min}(W_{1,0}) \leq n \sigma_{\min}(W_{1,0})$, condition (7) has the implication that

$$\sigma_{\min}(W_{1,0}) \geq n^{-r}. \tag{16}$$

Using (16), and $\|W_{1,0}\|_2 \leq 1$, $\|W_{1,t}\|_2 \leq 1$, we therefore get

$$\begin{aligned}
d_t &= \|W_{2,t}\|_2 \\
&\leq \|\Lambda_2^t\|_2 \cdot \|W_{2,0}\|_2 \cdot \|W_{1,0}^{-1}\|_2 \cdot \|\Lambda_1^{-t}\|_2 \cdot \|W_{1,t}\|_2 \\
&\leq \left( \frac{|\lambda_K|}{|\lambda_{K+1}|} \right)^{-t} n^r \\
&\leq \eta^{-t} n^r.
\end{aligned}$$

For any $s > 0$, if $t \geq ((s+r)\log n + \log 2)/\log \eta$, we have $d_t \leq n^{-s}/2$. $\quad\square$

To proceed with the analysis, we introduce two elementary but useful lemmas.

**Lemma 9.** *Suppose $Y \in \mathbb{R}^{n \times K}$ has the smallest singular value $\sigma_{\min}(Y) > 0$, where $K \leq n$. Let $y_k$ represent the $k$th column of $Y$. Then,*

$$\|y_k\| \geq \sigma_{\min}(Y),$$

*for all $1 \leq k \leq K$.*

*Proof.* Let $e_k \in \mathbb{R}^K$ be the elementary vector where the $k$th entry is 1 and all other entries are 0. Then $y_k = Y e_k$. The conclusion is clear by noting $\sigma_{\min}(Y) = \min_{\|x\|=1} \|Yx\|$. $\quad\square$

**Lemma 10.** *Suppose $Y \in \mathbb{R}^{n \times K}$ has the smallest singular value $\sigma_{\min}(Y) > 0$, where $K \leq n$. Let $y_k$ represent the $k$th column of $Y$. Assume the Gram-Schmidt process on $Y$ produces*

$$q_1 = \frac{y_1}{\|y_1\|},$$

$$u_k = y_k - \sum_{j=1}^{k-1} \langle y_k, q_j \rangle q_j, \quad 2 \leq k \leq K,$$

$$q_k = \frac{u_k}{\|u_k\|}, \quad 2 \leq k \leq K.$$

*Then*

$$\|u_k\| \geq \sigma_{\min}(Y)$$

*for all $1 \leq k \leq K$.*

*Proof.* From the Gram-Schmidt process, we know that $u_k$ can be expressed as $u_k = Y\varphi$, where $\varphi$ has 1 in its $k$th entry. This implies $\|\varphi\| \geq 1$. Then,

$$\|u_k\| = \|\varphi\| \cdot \|Y \frac{\varphi}{\|\varphi\|}\| \geq \|\varphi\| \cdot \sigma_{\min}(Y) \geq \sigma_{\min}(Y).$$

$\square$

If $|\lambda_K|$ is not too small, then throughout the iterations, $Y_t$ satisfies certain bounds uniformly, as demonstrated by Lemma 11. The condition on $|\lambda_K|$ will be discussed towards the end of this section.

**Lemma 11.** *Assume $Q_0$ satisfies (7). If $|\lambda_K| \geq \sqrt{2} n^{-(r-1)}$ holds, then for any $t \geq 0$, one has*

$$\sigma_{\min}(Y_{t+1}) \geq \frac{3}{4} n^{-(r-1)}. \tag{17}$$

$$\|y_{k,t+1}\| \leq n. \tag{18}$$

*Proof.* Observe that

$$Y_{t+1} = A Q_t = (V_1 \Lambda_1 V_1^\top + V_2 \Lambda_2 V_2^\top) Q_t = V_1 \Lambda_1 W_{1,t} + V_2 \Lambda_2 W_{2,t},$$

and that the columns of $V_1$ and $V_2$ are orthonormal, we have for any $x \in \mathbb{R}^K$,

$$\begin{aligned}
\|Y_{t+1} x\|^2 &= \|V_1 \Lambda_1 W_{1,t} x + V_2 \Lambda_2 W_{2,t} x\|^2 \\
&= \|V_1 \Lambda_1 W_{1,t} x\|^2 + \|V_2 \Lambda_2 W_{2,t} x\|^2 \\
&\geq \|V_1 \Lambda_1 W_{1,t} x\|^2 \\
&= \|\Lambda_1 W_{1,t} x\|^2.
\end{aligned}$$

Therefore, $\sigma_{\min}(Y_{t+1}) \geq \sigma_{\min}(\Lambda_1 W_{1,t})$. We next derive a lower-bound of $\sigma_{\min}(\Lambda_1 W_{1,t})$. From the relation (15), we know that $W_{2,t} (\Lambda_1 W_{1,t})^{-1} = \Lambda_2^t \left( W_{2,0} (\Lambda_1 W_{1,0})^{-1} \right) \Lambda_1^{-t}$. Hence,

$$\begin{aligned}
\|W_{2,t} (\Lambda_1 W_{1,t})^{-1}\|_2 &\leq \|\Lambda_2^t\|_2 \cdot \|W_{2,0}\|_2 \cdot \|(\Lambda_1 W_{1,0})^{-1}\|_2 \cdot \|\Lambda_1^{-t}\|_2 \\
&\leq \|(\Lambda_1 W_{1,0})^{-1}\|_2 \\
&= \sigma_{\min}^{-1}(\Lambda_1 W_{1,0}) \\
&\leq n^{r-1}.
\end{aligned}$$

The last inequality follows from (7). By the definitions of $W_{1,t}$ and $W_{2,t}$, we also have

$$W_{1,t}^\top W_{1,t} + W_{2,t}^\top W_{2,t} = Q_t^\top (V_1 V_1^\top + V_2 V_2^\top) Q_t = Q_t^\top Q_t = I_K.$$

Note that

$$\begin{aligned}
\left[ W_{2,t} (\Lambda_1 W_{1,t})^{-1} \right]^\top \left[ W_{2,t} (\Lambda_1 W_{1,t})^{-1} \right] &= \Lambda_1^{-1} W_{1,t}^{-\top} W_{2,t}^\top W_{2,t} W_{1,t}^{-1} \Lambda_1^{-1} \\
&= \Lambda_1^{-1} W_{1,t}^{-\top} (I_K - W_{1,t}^\top W_{1,t}) W_{1,t}^{-1} \Lambda_1^{-1} \\
&= \Lambda_1^{-1} (W_{1,t}^{-\top} W_{1,t}^{-1} - I_K) \Lambda_1^{-1} \\
&= (\Lambda_1 W_{1,t})^{-\top} (\Lambda_1 W_{1,t})^{-1} - \Lambda_1^{-2},
\end{aligned}$$

it then follows that

$$\|W_{2,t}\left(\Lambda_1 W_{1,t}\right)^{-1}\|_2^2 = \left\| \left[W_{2,t}\left(\Lambda_1 W_{1,t}\right)^{-1}\right]^\top \left[W_{2,t}\left(\Lambda_1 W_{1,t}\right)^{-1}\right] \right\|_2$$

$$\geq \|\left(\Lambda_1 W_{1,t}\right)^{-1}\|_2^2 - \|\Lambda_1^{-2}\|_2.$$

We have shown the left-hand-side is upper-bounded by $n^{2(r-1)}$ earlier, and we know that $\|\Lambda_1^{-2}\|_2 = \lambda_K^{-2} \leq n^{2(r-1)}/2$ holds. Therefore we get

$$\|\left(\Lambda_1 W_{1,t}\right)^{-1}\|_2^2 \leq \|W_{2,t}\left(\Lambda_1 W_{1,t}\right)^{-1}\|_2^2 + \lambda_K^{-2} \leq \frac{16}{9}n^{2(r-1)}.$$

This leads to

$$\sigma_{\min}\left(\Lambda_1 W_{1,t}\right) \geq \frac{3}{4}n^{-(r-1)}.$$

We therefore establish the following lower-bound of $\sigma_{\min}(Y_{t+1})$ uniformly for all $t \geq 0$:

$$\sigma_{\min}(Y_{t+1}) \geq \sigma_{\min}(\Lambda_1 W_{1,t}) \geq \frac{3}{4}n^{-(r-1)}.$$

Further, since $\|A\|_2 \leq n$, we have an uniform upper-bound of $\|y_{k,t+1}\|$ for all $t \geq 0, 1 \leq k \leq K$:

$$\|y_{k,t+1}\| = \|Aq_{k,t}\| \leq \|A\|_2\|q_{k,t}\| \leq n.$$

$\square$

**The GNN Approximation.** The GNN can be designed to emulate each step of orthogonal iteration method. It starts with $x^{(0)} = (\widehat{Q}_0, \mathbf{1}_n) \in \mathbb{R}^{n \times (K+1)}$, where $\widehat{Q}_0 = Q_0$ serves as the initial value of the GNN iterations. For the $t$-th iteration, the first procedure is to compute $\widehat{Y}_t = A\widehat{Q}_{t-1}$, which can be realized by one layer of GNN. The next procedure is the QR decomposition of $\widehat{Y}_t$. Let $\hat{y}_{k,t}$ represent the $k$th column of $\widehat{Y}_t$. We devise the architecture shown in Figure 1 to approximate the first step (12) of the QR decomposition. In this chart, we have suppressed the superscript of $\bar{z}$ in each layer for convenience. Also note that the full node feature is $x = (\rho(\bar{z}), \bar{z})$ in each layer, of which we omitted the first component $\rho(\bar{z})$ in the chart. We assume $m$ satisfy

$$m = \left\lceil \frac{2(s+r)r(\log n)^2}{\log \eta} + 2((K+1)r + s)\log n \right\rceil. \tag{19}$$

As illustrated in Figure 1, in the last layer of the first step, $\bar{z} = \left(\widehat{Y}_t, \hat{q}_{1,t}, \mathbf{1}_n\right)$ is produced. The next batch of GNN layers resumes from this layer, and tries to approximate the subsequent steps of QR decomposition. In general, assuming the GNN has generated $\bar{z} = \left(\widehat{Y}_t, \hat{q}_{1,t}, \ldots, \hat{q}_{k-1,t}, \mathbf{1}_n\right)$ in the last layer of the $(k-1)$th step of QR decomposition ($2 \leq k \leq K-1$), we design the GNN structure in Figure 2 to implement the $k$th step. Again the superscript of $\bar{z}$, that denotes which layer this node belongs to, is suppressed. After all $K$ steps of QR decomposition are carried out, we obtain $\widehat{Q}_t = [\hat{q}_{1,t}, \cdots, \hat{q}_{K,t}]$, which is then used for the $(t+1)$-th iteration.

We analyze how errors accumulate across layers and iterations in the designed GNN. Let $b_0 = n^r 2^{-m}$, $b_1 = 4n^r$, $b_2 = 49n^r$, $\mathbb{N}_0$ denote the set of all nonnegative integers, and

$$\mathbb{S}_{k,t} = \left\{ \mathbf{s} = (s_0, \ldots, s_k) : \sum_{j=0}^{k} s_j = t - 1, \text{ and } s_j \in \mathbb{N}_0 \text{ for all } 0 \leq j \leq k. \right\}$$

Then define

$$R_{k,0} = 0, \tag{20}$$

$$R_{k,t} = b_0 b_2^{k-1} \sum_{\mathbf{s} \in \mathbb{S}_{k,t}} \prod_{j=1}^{k} (b_1 j)^{s_j}, \quad 1 \leq t \leq T. \tag{21}$$

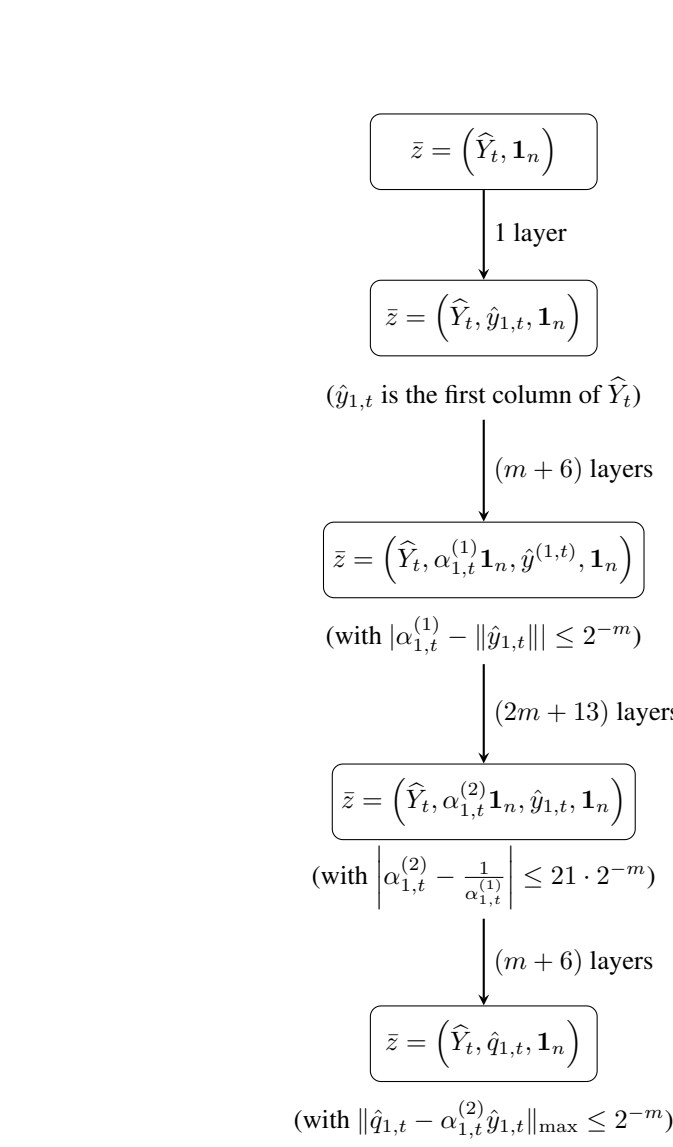

Figure 1: The GNN architecture to approximate the first step of QR decomposition.

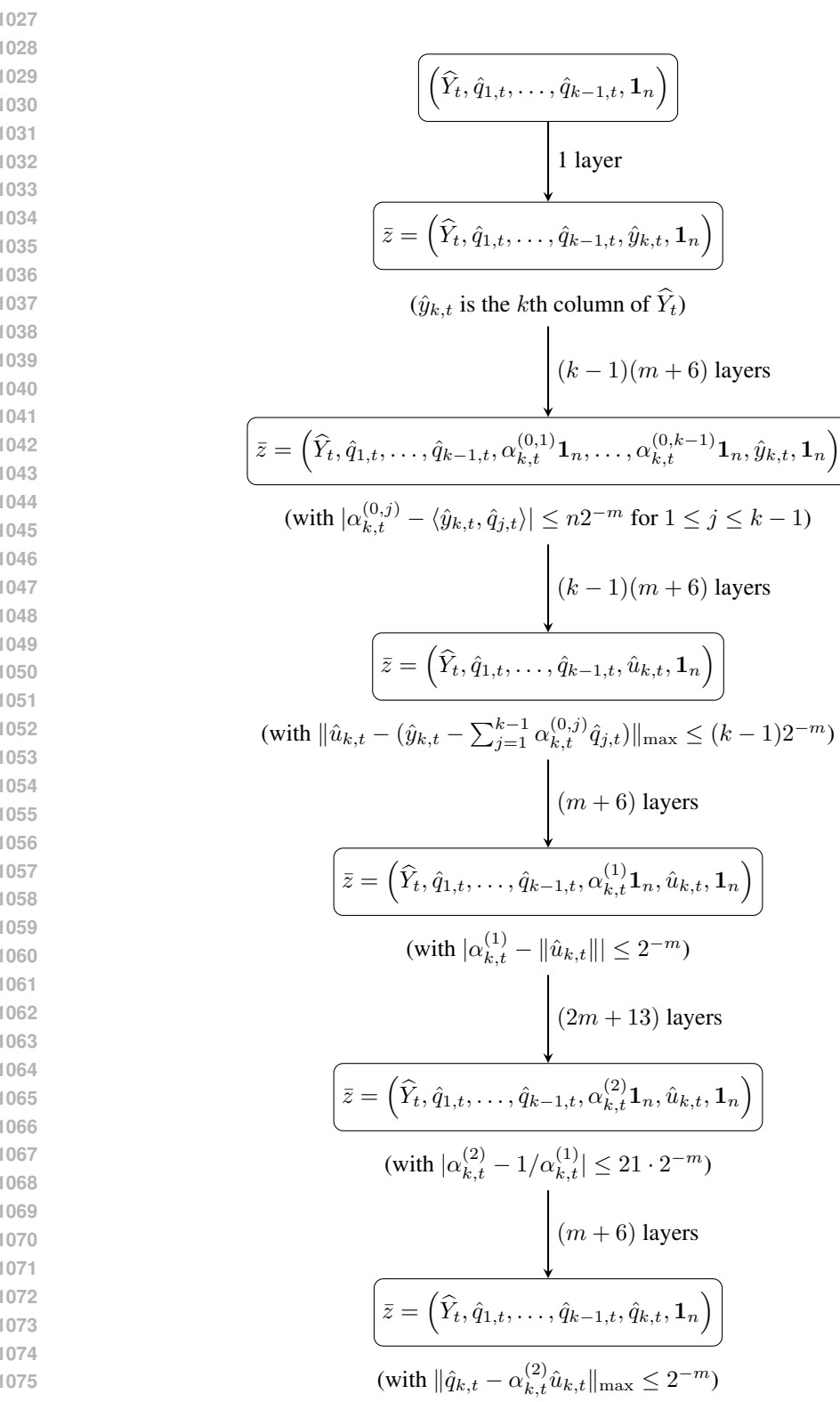

Figure 2: The GNN architecture to approximate the $k$th step of QR decomposition.

We adopt an induction to prove the following bound holds for $t = 0, 1, \ldots$ and all $1 \leq k \leq K$:

$$\|\hat{q}_{k,t} - q_{k,t}\| \leq R_{k,t}. \tag{22}$$

A few properties of $R_{k,t}$ that will be used in the induction are summarized in Lemma 12 and 13.

**Lemma 12.** *Suppose $R_{k,t}$ is defined by (20) and (20). Then,*

$$R_{k,t} = b_1 k R_{k,t-1} + b_2 R_{k-1,t} \tag{23}$$

*for $2 \leq k \leq K$ and $1 \leq t \leq T$. Moreover, $R_{k,t}$ is strictly increasing in both $t$ and $k$.*

*Proof.* $\mathbb{S}_{k,t}$ can be split into two subsets: $\mathbb{S}_{k,t}^{(1)}$ and $\mathbb{S}_{k,t}^{(2)}$, where elements of $\mathbb{S}_{k,t}^{(1)}$ satisfy $s_k \geq 1$ and elements of $\mathbb{S}_{k,t}^{(2)}$ satisfy $s_k = 0$. Elements of $\mathbb{S}_{k,t}^{(1)}$ have a one-to-one mapping $\mathbf{s} \to \mathbf{s}'$ to elements of $\mathbb{S}_{k,t-1}$ in the sense that, for any $\mathbf{s} = (s_0, \ldots, s_k) \in \mathbb{S}_{k,t}^{(1)}$, one has $\mathbf{s}' = (s_0, \ldots, s_k - 1) \in \mathbb{S}_{k,t-1}$. Therefore

$$\sum_{\mathbf{s} \in \mathbb{S}_{k,t}^{(1)}} \prod_{j=1}^{k} (b_1 j)^{s_j} = b_1 k \sum_{\mathbf{s} \in \mathbb{S}_{k,t-1}} \prod_{j=1}^{k} (b_1 j)^{s_j}.$$

On the other hand, elements of $\mathbb{S}_{k,t}^{(2)}$ have a one-to-one mapping to elements of $\mathbb{S}_{k-1,t}$ since $s_k = 0$. Then

$$\sum_{\mathbf{s} \in \mathbb{S}_{k,t}^{(2)}} \prod_{j=1}^{k} (b_1 j)^{s_j} = \sum_{\mathbf{s} \in \mathbb{S}_{k-1,t}} \prod_{j=1}^{k} (b_1 j)^{s_j}.$$

Combining the last two equalities, we get the desired result (23). $\square$

**Lemma 13.** *$R_{k,t}$, defined by (20) and (21), satisfies*

$$R_{k,t} \leq b_0 b_2^{k-1} t^k (k b_1)^{t-1}, \tag{24}$$

$$\sqrt{K} R_{K,T} \leq \frac{1}{2} n^{-s}, \tag{25}$$

$$(12 + K) R_{K,T} \leq \frac{1}{8} n^{-r}, \tag{26}$$

$$R_{k,t} \geq \sum_{j=1}^{k-1} R_{k-1,t}. \tag{27}$$

*Proof.* For any $\mathbf{s} \in \mathbb{S}_{k,t}$, we have

$$\prod_{j=1}^{k} (b_1 j)^{s_j} \leq \prod_{j=1}^{k} (b_1 k)^{s_j} = (b_1 k)^{\sum_{j=1}^{k} s_j} = (b_1 k)^{t-1-s_0}.$$

Plugging this into (21), we get

$$R_{k,t} \leq b_0 b_2^{k-1} \sum_{\mathbf{s} \in \mathbb{S}_{k,t}} (b_1 k)^{t-1-s_0}$$

$$= b_0 b_2^{k-1} \sum_{s_0=0}^{t-1} \binom{t+k-2-s_0}{k-1} (b_1 k)^{t-1-s_0}$$

$$= b_0 b_2^{k-1} \sum_{i=0}^{t-1} \binom{k+i-1}{k-1} (b_1 k)^{i}$$

$$\leq b_0 b_2^{k-1} \sum_{i=0}^{t-1} \binom{k+i-1}{k-1} (b_1 k)^{t-1}$$

$$= b_0 b_2^{k-1} \binom{t+k-1}{k} (b_1 k)^{t-1}.$$

From the inequality $\binom{t+k-1}{k} \leq t^k$, (24) is established.

From (24), we know that

$$\log\left(\sqrt{K}R_{K,T}\right) - \log\left(\frac{1}{2}n^{-s}\right)$$

$$\leq \frac{1}{2}\log K + \log b_0 + (K-1)\log b_2 + K\log T + (T-1)(\log K + \log b_1) + \log 2 + s\log n$$

$$\leq \frac{1}{2}\log K + r\log n - m\log 2 + (K-1)(\log 49 + r\log n) + K\log\left[\frac{(s+r)\log n + \log 2}{\log \eta} + 1\right]$$

$$+ \left[\frac{(s+r)\log n + \log 2}{\log \eta}\right](\log K + \log 4 + r\log n) + \log 2 + s\log n$$

$$\leq -m\log 2 + \frac{(2\log 2)(s+r)r(\log n)^2}{\log \eta} + (2\log 2)(Kr+s)\log n.$$

The choice of $m$ in (19) guarantees that

$$m\log 2 \geq \left[\frac{2(s+r)r(\log n)^2}{\log \eta} + 2((K+1)r+s)\log n\right]\log 2$$

$$\geq \frac{(2\log 2)(s+r)r(\log n)^2}{\log \eta} + (2\log 2)(Kr+s)\log n.$$

So (25) is proved.

Similarly, we have

$$\log\left((12+K)R_{K,T}\right) - \log\left(\frac{1}{8}n^{-r}\right)$$

$$\leq \log(12+K) + \log b_0 + (K-1)\log b_2 + K\log T + (T-1)(\log K + \log b_1) + \log 8 + r\log n$$

$$\leq \log(12+K) + r\log n - m\log 2 + (K-1)(\log 49 + r\log n) + K\log\left[\frac{(s+r)\log n + \log 2}{\log \eta} + 1\right]$$

$$+ \left[\frac{(s+r)\log n + \log 2}{\log \eta}\right](\log K + \log 4 + r\log n) + \log 8 + r\log n$$

$$\leq -m\log 2 + \frac{(2\log 2)(s+r)r(\log n)^2}{\log \eta} + (2\log 2)(K+1)r\log n$$

$$\leq -\left[\frac{2(s+r)r(\log n)^2}{\log \eta} + 2((K+1)r+s)\log n\right]\log 2$$

$$+ \frac{(2\log 2)(s+r)r(\log n)^2}{\log \eta} + (2\log 2)(K+1)r\log n$$

$$\leq 0.$$

Thus (26) also holds.

Finally, by (23) we know that $R_{k,t} \geq b_2 R_{k-1,t}$. For a fixed $t$, the value changes of $R_{k,t}$ along the direction of $k$ is faster than a geometric sequence with common ratio $b_2$. Then (27) is valid.

$$\square$$

To start with the induction, for $t=0$, we have $\widehat{Q}_0 = Q_0$, so (22) holds. Now assume (22) holds for $0, \ldots, t-1$ with $t \geq 1$. For $t$, we first have $\widehat{Y}_t = A\widehat{Q}_{t-1}$. Based on $\|\hat{q}_{k,t-1} - q_{k,t-1}\| \leq R_{k,t-1}$, we can immediately obtain the following bounds:

$$\|\hat{y}_{k,t} - y_{k,t}\| \leq nR_{k,t-1}, \tag{28}$$

$$\|\hat{q}_{k,t-1}\| \leq 2, \tag{29}$$

$$\|\hat{y}_{k,t}\| \leq 2n, \tag{30}$$

$$\sigma_{\min}(\widehat{Y}_t) \geq \frac{5}{8}n^{-(r-1)}, \tag{31}$$

In particular, (28) holds because $\|\hat{y}_{k,t} - y_{k,t}\| = \|A(\hat{q}_{k,t-1} - q_{k,t-1}\| \leq \|A\|_2\|\hat{q}_{k,t-1} - q_{k,t-1}\| \leq n\|\hat{q}_{k,t-1} - q_{k,t-1}\|$. As $R_{k,t-1} \leq 1$, we get $\|\hat{q}_{k,t-1}\| \leq \|\hat{q}_{k,t-1} - q_{k,t-1}\| + \|q_{k,t-1}\| \leq 2$. Moreover, using (18), we know that $\|\hat{y}_{k,t}\| \leq \|\hat{y}_{k,t} - y_{k,t}\| + \|y_{k,t}\| \leq nR_{k,t-1} + n \leq 2n$. Finally, by Weyl's inequality for singular values, we get

$$|\sigma_{\min}(\widehat{Y}_t) - \sigma_{\min}(Y_t)| \leq \|\widehat{Y}_t - Y_t\|_2 \leq \sqrt{K} \max_{1 \leq k \leq K} \|\hat{y}_{k,t} - y_{k,t}\| \leq \sqrt{K}nR_{K,t-1} \leq \frac{1}{8}n^{-(r-1)}.$$

The last inequality is a consequence of (26) in Lemma 13 and the fact that $\sqrt{K} \leq 12 + K$. In view of (17), we have

$$|\sigma_{\min}(\widehat{Y}_t)| \geq |\sigma_{\min}(Y_t)| - |\sigma_{\min}(\widehat{Y}_t) - \sigma_{\min}(Y_t)| \geq \frac{3}{4}n^{-(r-1)} - \frac{1}{8}n^{-(r-1)} = \frac{5}{8}n^{-(r-1)}.$$

By the GNN structure in Figure 1, we get

$$\left\|\hat{q}_{1,t} - \frac{\hat{y}_{1,t}}{\|\hat{y}_{1,t}\|}\right\| \leq \|\hat{q}_{1,t} - \alpha_{1,t}^{(2)}\hat{y}_{1,t}\| + \left|\alpha_{1,t}^{(2)} - \frac{1}{\alpha_{1,t}^{(1)}}\right|\|\hat{y}_{1,t}\| + \left|\frac{1}{\alpha_{1,t}^{(1)}} - \frac{1}{\|\hat{y}_{1,t}\|}\right|\|\hat{y}_{1,t}\|$$

$$= \|\hat{q}_{1,t} - \alpha_{1,t}^{(2)}\hat{y}_{1,t}\| + \left|\alpha_{1,t}^{(2)} - \frac{1}{\alpha_{1,t}^{(1)}}\right|\|\hat{y}_{1,t}\| + \left|\frac{\alpha_{1,t}^{(1)} - \|\hat{y}_{1,t}\|}{\alpha_{1,t}^{(1)}}\right|$$

The bound (30) directly suggests that $\|\hat{y}_{1,t}\| \leq 2n$. From (31) and Lemma 9, we also know that $\|\hat{y}_{1,t}\| \geq \sigma_{\min}(\widehat{Y}_t) \geq 5n^{-(r-1)}/8$. Then $|\alpha_{1,t}^{(1)}| \geq \|\hat{y}_{1,t}\| - |\alpha_{1,t}^{(1)} - \|\hat{y}_{1,t}\|| \geq 5n^{-(r-1)}/8 - 2^{-m} \geq n^{-(r-1)}/2$, since $2^{-m} \leq n^{-(r-1)}/8$ by (19). Therefore,

$$\left\|\hat{q}_{1,t} - \frac{\hat{y}_{1,t}}{\|\hat{y}_{1,t}\|}\right\| \leq 2^{-m} + 21 \cdot 2^{-m} \cdot 2n + \frac{2^{-m}}{n^{-(r-1)}/2} \leq n^r 2^{-m} = b_0.$$

Next we derive an upper bound for $\|\hat{q}_{1,t} - q_{1,t}\|$. The triangle inequality implies

$$\|\hat{q}_{1,t} - q_{1,t}\| \leq \left\|\hat{q}_{1,t} - \frac{\hat{y}_{1,t}}{\|\hat{y}_{1,t}\|}\right\| + \left\|\frac{\hat{y}_{1,t}}{\|\hat{y}_{1,t}\|} - \frac{y_{1,t}}{\|\hat{y}_{1,t}\|}\right\| + \left\|\frac{y_{1,t}}{\|\hat{y}_{1,t}\|} - \frac{y_{1,t}}{\|y_{1,t}\|}\right\|$$

$$\leq b_0 + \frac{1}{\|\hat{y}_{1,t}\|}\|\hat{y}_{1,t} - y_{1,t}\| + \frac{|\|y_{1,t}\| - \|\hat{y}_{1,t}\||}{\|\hat{y}_{1,t}\|}.$$

According to (28) and (31), the second and third terms in the display are controlled by

$$\frac{|\|y_{1,t}\| - \|\hat{y}_{1,t}\||}{\|\hat{y}_{1,t}\|} \leq \frac{1}{\|\hat{y}_{1,t}\|}\|\hat{y}_{1,t} - y_{1,t}\| \leq \frac{nR_{1,t-1}}{n^{-(r-1)}/2} = 2n^r R_{1,t-1}.$$

Hence, we obtain

$$\|\hat{q}_{1,t} - q_{1,t}\| \leq b_0 + 4n^r R_{1,t-1} = b_0 + b_1 R_{1,t-1}.$$

Note that $R_{1,t} = b_0 \sum_{s_1=0}^{t-1} b_2^{s_1} = b_0(b_1^t - 1)/(b_1 - 1)$ by its definition (21), the previous display leads to

$$\|\hat{q}_{1,t} - q_{1,t}\| \leq b_0 + b_1 b_0 \frac{b_1^{t-1} - 1}{b_1 - 1} = b_0 \frac{b_1^t - 1}{b_1 - 1} = R_{1,t}.$$

Thus we have proved (22) for $t$ and $k = 1$.

Next we apply an inner induction on $k$ with the current $t$. Assume (22) holds for the current $t$ and $1, \ldots, k-1$. We first have for $1 \leq j \leq k-1$

$$|\langle \hat{y}_{k,t}, \hat{q}_{j,t}\rangle - \langle y_{k,t}, q_{j,t}\rangle| \leq |\langle \hat{y}_{k,t}, \hat{q}_{j,t} - q_{j,t}\rangle| + |\langle \hat{y}_{k,t} - y_{k,t}, q_{j,t}\rangle|$$

$$\leq \|\hat{y}_{k,t}\| \cdot \|\hat{q}_{j,t} - q_{j,t}\| + \|\hat{y}_{k,t} - y_{k,t}\| \cdot \|q_{j,t}\|$$

$$\leq 2nR_{j,t} + nR_{k,t-1}, \tag{32}$$

where in the last inequality we have used (22), (28) and (30).

Let $\tilde{u}_{k,t} = \hat{y}_{k,t} - \sum_{j=1}^{k-1}\langle \hat{y}_{k,t}, \hat{q}_{j,t}\rangle \hat{q}_{j,t}$. From (32) and the inequalities (28) and (29), we have

$$\|\tilde{u}_{k,t} - u_{k,t}\| = \|\hat{y}_{k,t} - \sum_{j=1}^{k-1}\langle \hat{y}_{k,t}, \hat{q}_{j,t}\rangle \hat{q}_{j,t} - y_{k,t} + \sum_{j=1}^{k-1}\langle y_{k,t}, q_{j,t}\rangle q_{j,t}\|$$

$$\leq \|\hat{y}_{k,t} - y_{k,t}\| + \sum_{j=1}^{k-1}|\langle \hat{y}_{k,t}, \hat{q}_{j,t}\rangle|\|\hat{q}_{j,t} - q_{j,t}\| + \sum_{j=1}^{k-1}|\langle \hat{y}_{k,t}, \hat{q}_{j,t}\rangle - \langle y_{k,t}, q_{j,t}\rangle|\|q_{j,t}\|$$

$$\leq nR_{k,t-1} + \sum_{j=1}^{k-1}\|\hat{y}_{k,t}\| \cdot \|\hat{q}_{j,t}\| \cdot \|\hat{q}_{j,t} - q_{j,t}\| + \sum_{j=1}^{k-1}(2nR_{j,t} + nR_{k,t-1})$$

$$\leq nR_{k,t-1} + \sum_{j=1}^{k-1}4nR_{j,t} + \sum_{j=1}^{k-1}(2nR_{j,t} + nR_{k,t-1})$$

$$= 6n\sum_{j=1}^{k-1}R_{j,t} + knR_{k,t-1}$$

$$\leq 12nR_{k-1,t} + knR_{k,t-1}, \tag{33}$$

where the last inequality holds because $\sum_{j=1}^{k-1}R_{j,t} \leq 2R_{k-1,t}$ according to (27).

In view of (26) in Lemma 13, $12nR_{k-1,t} + knR_{k,t-1} \leq (12+K)nR_{K,T} \leq n^{-(r-1)}/8$. We thereby have that

$$\|\tilde{u}_{k,t} - u_{k,t}\| \leq n^{-(r-1)}/8. \tag{34}$$

By Lemma 10 and (17), we know that $\|u_{k,t}\| \geq \sigma_{\min}(Y_t) \geq 5n^{-(r-1)}/8$. By (34), we get

$$\|\tilde{u}_{k,t}\| \geq \|u_{k,t}\| - \|\tilde{u}_{k,t} - u_{k,t}\| \geq \frac{5}{8}n^{-(r-1)} - \frac{1}{8}n^{-(r-1)} = \frac{1}{2}n^{-(r-1)}. \tag{35}$$

Meantime, since $12nR_{k-1,t} + knR_{k,t-1} \leq n$, from (18) and (34), we also get

$$\|\tilde{u}_{k,t}\| \leq \|u_{k,t}\| + \|\tilde{u}_{k,t} - u_{k,t}\| \leq \|y_{k,t}\| + \|\tilde{u}_{k,t} - u_{k,t}\| \leq n + n = 2n. \tag{36}$$

By the GNN structure in Figure 2, we have

$$\|\hat{u}_{k,t} - \tilde{u}_{k,t}\| \leq \|\hat{u}_{k,t} - (\hat{y}_{k,t} - \sum_{j=1}^{k-1}\alpha_{k,t}^{(0,j)}\hat{q}_{j,t})\| + \sum_{j=1}^{k-1}|\alpha_{k,t}^{(0,j)} - \langle \hat{y}_{k,t}, \hat{q}_{j,t}\rangle| \cdot \|\hat{q}_{j,t}\|$$

$$\leq \sqrt{n}\|\hat{u}_{k,t} - (\hat{y}_{k,t} - \sum_{j=1}^{k-1}\alpha_{k,t}^{(0,j)}\hat{q}_{j,t})\|_{\max} + \sum_{j=1}^{k-1}|\alpha_{k,t}^{(0,j)} - \langle \hat{y}_{k,t}, \hat{q}_{j,t}\rangle| \cdot \|\hat{q}_{j,t}\|$$

$$\leq (k-1)2^{-m}\sqrt{n} + 2(k-1)n2^{-m}$$

$$\leq 2kn2^{-m}. \tag{37}$$

Observe that $m > r\log n/\log 2$ by condition (19), thus $2kn2^{-m} \leq 2kn^{-r} \leq n^{-(r-1)}/4$. It then follows that

$$\|\hat{u}_{k,t}\| \geq \|\tilde{u}_{k,t}\| - \|\hat{u}_{k,t} - \tilde{u}_{k,t}\| \geq \frac{1}{2}n^{-(r-1)} - \frac{1}{4}n^{-(r-1)} = \frac{1}{4}n^{-(r-1)}. \tag{38}$$

Also notice that $2kn2^{-m} \leq n$, it follows that

$$\|\hat{u}_{k,t}\| \leq \|\tilde{u}_{k,t}\| + \|\tilde{u}_{k,t} - \hat{u}_{k,t}\| \leq 2n + n = 3n. \tag{39}$$

We also have

$$|\alpha_{k,t}^{(1)}| \geq \|\hat{u}^{((k,t)}\| - |\alpha_{k,t}^{(1)} - \|\hat{u}^{((k,t)}\|| \geq \frac{1}{4}n^{-(r-1)} - 2^{-m} \geq \frac{1}{8}n^{-(r-1)}. \tag{40}$$

where the last inequality is implied by $2^{-m} \leq n^{-(r-1)}/8$, which can be derived from (19).

Using (35), (36), (37) and (38), we can further get

$$
\begin{aligned}
\left\| \frac{\hat{u}_{k,t}}{\|\hat{u}_{k,t}\|} - \frac{\tilde{u}_{k,t}}{\|\tilde{u}_{k,t}\|} \right\| &\leq \frac{1}{\|\hat{u}_{k,t}\|}\|\hat{u}_{k,t} - \tilde{u}_{k,t}\| + \frac{|\|\tilde{u}_{k,t}\| - \|\hat{u}_{k,t}\||}{\|\hat{u}_{k,t}\| \cdot \|\tilde{u}_{k,t}\|} \cdot \|\tilde{u}_{k,t}\| \\
&= \frac{1}{\|\hat{u}_{k,t}\|}\|\hat{u}_{k,t} - \tilde{u}_{k,t}\| + \frac{|\|\tilde{u}_{k,t}\| - \|\hat{u}_{k,t}\||}{\|\hat{u}_{k,t}\|} \\
&\leq \frac{2kn2^{-m}}{n^{-(r-1)/4}} + \frac{2kn2^{-m}}{n^{-(r-1)/4}} \\
&\leq 16kn^r 2^{-m}.
\end{aligned}
\tag{41}
$$

Combining (39), (40), (41), we have

$$
\begin{aligned}
\left\| \hat{q}^{(k,t)} - \frac{\tilde{u}_{k,t}}{\|\tilde{u}_{k,t}\|} \right\| &\leq \left\| \hat{q}^{(k,t)} - \alpha_{k,t}^{(2)}\hat{u}_{k,t} \right\| + \left\| (\alpha_{k,t}^{(2)} - 1/\alpha_{k,t}^{(1)})\hat{u}_{k,t} \right\| \\
&\quad + \left\| \left( \frac{1}{\alpha_{k,t}^{(1)}} - \frac{1}{\|\hat{u}^{((k,t)}\|} \right) \hat{u}_{k,t} \right\| + \left\| \frac{\hat{u}_{k,t}}{\|\hat{u}_{k,t}\|} - \frac{\tilde{u}_{k,t}}{\|\tilde{u}_{k,t}\|} \right\| \\
&\leq \sqrt{n}\left\| \hat{q}^{(k,t)} - \alpha_{k,t}^{(2)}\hat{u}_{k,t} \right\|_{\max} + |\alpha_{k,t}^{(2)} - 1/\alpha_{k,t}^{(1)}| \cdot \|\hat{u}_{k,t}\| \\
&\quad + \left| \frac{\alpha_{k,t}^{(1)} - \|\hat{u}_{k,t}\|}{\alpha_{k,t}^{(1)}} \right| + \left\| \frac{\hat{u}_{k,t}}{\|\hat{u}_{k,t}\|} - \frac{\tilde{u}_{k,t}}{\|\tilde{u}_{k,t}\|} \right\| \\
&\leq \sqrt{n} \cdot 2^{-m} + (21 \cdot 2^{-m})3n + 8n^{r-1} \cdot 2^{-m} + 16kn^r 2^{-m} \\
&\leq 17kn^r 2^{-m}.
\end{aligned}
$$

Now we are ready to bound $\|\hat{q}^{(k,t)} - q^{(k,t)}\|$. Note that $q^{(k,t)} = u^{(k,t)}/\|u^{(k,t)}\|$, so

$$
\begin{aligned}
\|\hat{q}^{(k,t)} - q^{(k,t)}\| &\leq \left\| \hat{q}^{(k,t)} - \frac{\tilde{u}_{k,t}}{\|\tilde{u}_{k,t}\|} \right\| + \frac{1}{\|\tilde{u}_{k,t}\|}\|\tilde{u}_{k,t} - u_{k,t}\| + \frac{|\|u_{k,t}\| - \|\tilde{u}_{k,t}\||}{\|u_{k,t}\| \cdot \|\tilde{u}_{k,t}\|} \cdot \|u_{k,t}\| \\
&= \left\| \hat{q}^{(k,t)} - \frac{\tilde{u}_{k,t}}{\|\tilde{u}_{k,t}\|} \right\| + \frac{1}{\|\tilde{u}_{k,t}\|}\|\tilde{u}_{k,t} - u_{k,t}\| + \frac{|\|u_{k,t}\| - \|\tilde{u}_{k,t}\||}{\|\tilde{u}_{k,t}\|} \\
&\leq 17kn^r 2^{-m} + \frac{12nR_{k-1,t} + knR_{k,t-1}}{n^{-(r-1)/2}} + \frac{12nR_{k-1,t} + knR_{k,t-1}}{n^{-(r-1)/2}} \\
&= 17kn^r 2^{-m} + 4kn^r R_{k,t-1} + 48n^r R_{k-1,t} \\
&\leq 4kn^r R_{k,t-1} + 49n^r R_{k-1,t} = R_{k,t},
\end{aligned}
$$

where the last inequality holds because $17kn^r 2^{-m} \leq n^{2r}2^{-m} = n^r R_{1,1} \leq n^r R_{k-1,t}$, and the last equality is due to Lemma 12. Therefore, (22) is established for $t$ and $k$. Both the inductions on $k$ and the induction on $t$ are now complete.

We now derive a bound for $\hat{d}_T := \text{dist}(\text{col}(\widehat{Q}_T), \text{col}(V_1)) = \|V_2^\top \widehat{Q}_T\|_2$. Using (22) and (25), we can get

$$
\|\hat{q}_{k,T} - q_{k,T}\| \leq \frac{n^{-s}}{2\sqrt{K}}. \quad k = 1, \ldots, K.
\tag{42}
$$

Therefore,

$$
\|\widehat{Q}_T - Q_T\|_2 \leq \sqrt{K} \max_{1 \leq k \leq K} \|q_{k,T} - q_{k,T}\| \leq \sqrt{K}R_{K,T} \leq \frac{1}{2}n^{-s}.
$$

Then by Lemma 8, we have

$$
\hat{d}_T \leq d_T + |\hat{d}_T - d_T| \leq \frac{1}{2}n^{-s} + \|V_2^\top(\widehat{Q}_T - Q_T)\|_2 \leq \frac{1}{2}n^{-s} + \|\widehat{Q}_T - Q_T\|_2 \leq n^{-s}.
\tag{43}
$$

The first result in Theorem 1 is established by letting $\widehat{Q} = \widehat{Q}_T$.

According to Figures 1 and 2, the number of layers needed in GNN for the $k$th step in the $t$-th orthogonal iteration is $(2k + 2)m + 12k + 14$. Also count in the layer to produce $\widehat{Y}_t = A\widehat{Q}_{t-1}$ in each iteration. So the total number of layers is at most

$$\sum_{k=1}^{K} \left((2k + 2)m + 12k + 14\right) T_k + T_k$$

$$= \left[K(K + 1)(m + 6) + 2K(m + 7) + 1\right] T$$

$$\leq 2K^2 m T_k$$

$$\leq 2K^2 \left[\frac{2(s + r)r(\log n)^2}{\log \eta} + 2((K + 1)r + s)\log n\right] \left[\frac{(s + r)\log n + \log 2}{\log \eta}\right]$$

$$\leq 8K^2(s + r)^2 r \frac{(\log n)^3}{(\log \eta)^2} + 8K^2((K + 1)r + s)(s + r)\frac{(\log n)^2}{\log \eta}.$$

**High probability bounds concerning eigenvalues of $A$.** The eigenvalues $\lambda_K$ and $\lambda_{K+1}$ of $A$ affects the approximation results of GNN. Specifically, condition $|\lambda_K| \geq \sqrt{2}n^{-(r-1)}$ required in Lemma 11, and the total number of layers depends on $\eta = |\lambda_K|/|\lambda_{K+1}|$. We provide high-probability bounds on $|\lambda_K|$ and $\eta$.

Let $\gamma_1, \ldots, \gamma_K$ be the first $K$ eigenvectors of $P$ with $|\gamma_1| \geq \cdots \geq |\gamma_K|$, and $u_1, \ldots, u_K$ be associated eigenvectors. Denote $U = [u_1, \ldots, u_K]$ and $\Gamma = \mathrm{diag}(\gamma_1, \ldots, \gamma_K)$. Define $P_0 = (p - q)I_K + q\mathbf{1}_K\mathbf{1}_K^\top \in \mathbb{R}^{K \times K}$ to be a "collapsed" version of $P$, and $N = \mathrm{diag}(n_1, \ldots, n_K)$. Further, let $Z \in \mathbb{R}^{n \times K}$ be the one-hot matrix of the true community labels. That is, $Z_{i,k} = \mathbb{1}_{\{\sigma_i = k\}}$ for $i \in [n]$, $k \in [K]$. We first state several preliminary lemmas.

**Lemma 14.** *The first $K$ eigenvalues $\gamma_1, \ldots, \gamma_K$ of $P$ are equal to the eigenvalues of $G = N^{1/2}P_0N^{1/2}$.*

*Proof.* First we have the equalities $P = ZP_0Z^\top$ and $Z^\top Z = N$. Assume nonzero vector $x \in \mathbb{R}^K$ satisfies $Gx = \gamma x$. Define $y = ZN^{-1/2}x \in \mathbb{R}^n$. Then

$$Py = ZP_0Z^\top ZN^{-1/2}x = ZP_0N^{1/2}x = ZN^{-1/2}Gx = \gamma ZN^{-1/2}x = \gamma y.$$

On the other hand, suppose nonzero vector $y \in \mathbb{R}^n$ satisfies $Py = \gamma y$. Let $x = N^{-1/2}Z^\top y \in \mathbb{R}^K$. Then

$$Gx = N^{1/2}P_0Z^\top y = N^{-1/2}Z^\top ZP_0Z^\top y = N^{-1/2}Z^\top Py = \gamma N^{-1/2}Z^\top y = \gamma x.$$

This completes the proof. $\qquad\square$

**Lemma 15.** *Let $G = N^{1/2}P_0N^{1/2}$. Then we have*

$$\sigma_{\min}(G) \geq (p - q)n_{\min}.$$

*The equality holds when $n_1 = \cdots = n_K$.*

*Proof.* We first write $G = (p - q)N + q\psi\psi^\top$, where $\varphi = (\sqrt{n_1}, \ldots, \sqrt{n_K})^\top$. For any vector $x \in \mathbb{R}^K$, we have

$$Gx = (p - q)x^\top Nx + q(x^\top \varphi)^2 \geq (p - q)x^\top Nx \geq (p - q)n_{\min}.$$

Hence the conclusion holds. The last statement follows by direct calculation. $\qquad\square$

The following result on spectral bound is essentially Theorem 5.2 in Lei & Rinaldo (2015) with a slightly different statement:

**Lemma 16** ((Lei & Rinaldo, 2015))**.** *For any $c_0 > 0$, there exists a constant $c_1$ that depends on $c_0$ such that*

$$\|A - P\|_2 \leq c_1\sqrt{np + \log n}$$

*with probability at least $1 - n^{-c_0}$.*

In view of Lemma 16, we constrain the analysis in the event that $\|A - P\|_2 \leq c_1\sqrt{np + \log n}$. By Weyl's inequality,

$$|\lambda_K - \gamma_K| \leq \|A - P\|_2.$$

Combining Lemmas 14 and 15, we have that

$$|\lambda_K| \geq |\gamma_K| - \|A - P\|_2 \geq (p - q)n_{\min} - c_1\sqrt{np + \log n} \geq c_2 n(p - q) \geq \sqrt{\log n}.$$

The last two inequalities holds because $n_{\min} \asymp n$, $n(p - q) \gg \sqrt{np}$, and $n(p - q) \gg \sqrt{\log n}$, from Assumptions 1 and 2. We get $|\lambda_K| \geq \sqrt{2}n^{-(r-1)}$ as required.

To derive a bound of $\log \eta$, observe that the $(K + 1)$th eigenvalue of $P$ is 0. By Weyl's inequality, we have

$$|\lambda_{K+1}| \leq \|A - P\|_2 \leq c_1\sqrt{np + \log n}.$$

Then it holds that

$$\log \eta \geq \log\left(\frac{(p - q)n_{\min}}{c_1\sqrt{np + \log n}} - 1\right) \geq \log\left(\frac{c_2 n(p - q)}{\max\{\sqrt{np}, \sqrt{\log n}\}}\right) = \xi.$$

This concludes the proof of Theorem 1.

## D  AN INITIALIZATION PROCEDURE

A natural way of getting the initial features $Q_0$ is to draw from the Haar distribution. Suppose $S_0 \in \mathbb{R}^{n \times K}$ is a random matrix where its entries are i.i.d. $N(0, 1)$. Let its QR decomposition obtained from the Gram-Schmidt process be $S_0 = Q_0 R_0$. Then $Q_0$ is Haar-distributed on the orthogonal group.

We first have the following result.

**Lemma 17.** *For a given A, $Q_0$ satisfies*

$$\mathbb{P}\left(\sigma_{\min}(\Lambda_1 V_1^\top Q_0) \geq \frac{|\lambda_K|n^{-(c_0+1/2)}}{(1 + \delta)\sqrt{K}}\right) \geq 1 - 2n^{-c_0}$$

*for any $\delta > 0$ and any $c_0 > 0$.*

*Proof.* By definition, we have $S_0^\top S_0 = R_0^\top R_0$. Then $\|R_0\|_2 = \|S_0\|_2$. Following the result in Rudelson & Vershynin (2011) with respect to the largest singular value, we have

$$\mathbb{P}\left(\|S_0\|_2 > \sqrt{n} + \sqrt{K} + t\right) \leq 2e^{-t^2/2}, \quad t > 0.$$

For an absolute constant $c_0$, take $t = \sqrt{2\log 2 + 2c_0 \log n}$. The probability bound of the last display becomes $n^{-c_0}$. Since $\sqrt{n} + \sqrt{K} + t \leq (1 + \delta)\sqrt{n}$ for any $\delta > 0$, we get

$$\mathbb{P}\left(\|S_0\|_2 > (1 + \delta)\sqrt{n}\right) \leq \mathbb{P}\left(\|S_0\|_2 > \sqrt{n} + \sqrt{K} + t\right) \leq n^{-c_0}$$

for any $\delta > 0$, with $n$ large enough. Therefore, we obtain

$$\mathbb{P}\left(\sigma_{\min}(R_0^{-1}) < \frac{1}{(1 + \delta)\sqrt{n}}\right) \leq n^{-c_0}.$$

Since columns of $V_1$ are all orthonormal, $V_1^\top S_0 \in \mathbb{R}^{K \times K}$ has i.i.d. $N(0, 1)$ entries. Following the result in Rudelson & Vershynin (2011) on the smallest singular value, we have

$$\mathbb{P}\left(\sigma_{\min}(V_1^\top S_0) < \frac{\epsilon}{\sqrt{K}}\right) \leq \epsilon.$$

We take $\epsilon = n^{-c_0}$.

Note $V_1^\top Q_0 = V_1^\top S_0 R_0^{-1}$. When $\sigma_{\min}(V_1^\top S_0)$ and $\sigma_{\min}(R_0^{-1})$ are both bounded away from 0, we have $\sigma_{\min}(V_1^\top Q_0) \geq \sigma_{\min}(V_1^\top S_0) \cdot \sigma_{\min}(R_0^{-1})$. It then follows that

$$
\mathbb{P}\left(\sigma_{\min}(\Lambda_1 V_1^\top Q_0) \geq \frac{|\lambda_K| n^{-(c_0+1/2)}}{(1+\delta)\sqrt{K}}\right)
$$

$$
\geq \mathbb{P}\left(\sigma_{\min}(V_1^\top Q_0) \geq \frac{n^{-c_0}}{\sqrt{K}} \text{ and } \sigma_{\min}(R_0^{-1}) \geq \frac{1}{(1+\delta)\sqrt{n}}\right)
$$

$$
\geq 1 - \mathbb{P}\left(\sigma_{\min}(V_1^\top Q_0) < \frac{n^{-c_0}}{\sqrt{K}}\right) - \mathbb{P}\left(\sigma_{\min}(R_0^{-1}) < \frac{1}{(1+\delta)\sqrt{n}}\right)
$$

$$
\geq 1 - 2n^{-c_0}.
$$

$\square$

The lower bound given in Lemma 17 still depends on $|\lambda_K|$. Lemma 18 further assures that $Q_0$ satisfies (7) with high probability.

**Lemma 18.** *If $Q_0 \in \mathbb{R}^{n \times K}$ is Haar distributed on the orthogonal group, then*

$$
\mathbb{P}\left(\sigma_{\min}(\Lambda_1 V_1^\top Q_0) \geq n^{-(c_0+1/2)}\right) \geq 1 - 3n^{-c_0}
$$

*for any $c_0 > 0$.*

*Proof.* The magnitude of $\lambda_K$ is analyzed at the end of Appendix C. Using Assumptions 1 and 2, we have that, with probability at least $1 - n^{-c_0}$,

$$
|\lambda_K| \geq |\gamma_K| - \|A - P\|_2 \geq (p-q)n_{\min} - c_1\sqrt{np + \log n} \geq c_2 n(p-q) \geq c_2\sqrt{\log n}
$$

for some $c_2 > 0$.

In view of with Lemma 17, we get that with probability at least $1 - 3n^{-c_0}$,

$$
\sigma_{\min}(\Lambda_1 V_1^\top Q_0) \geq \frac{|\lambda_K| n^{-(c_0+1/2)}}{(1+\delta)\sqrt{K}} \geq \frac{c_2\sqrt{\log n} \cdot n^{-(c_0+1/2)}}{(1+\delta)\sqrt{K}} \geq n^{-(c_0+1/2)}.
$$

$\square$

# E    PROOF OF THEOREM 2

We have the equality $PZ = ZP_0 N$, or equivalently, $PZN^{-1}P_0^{-1} = Z$. As $P = U\Gamma U^\top$, by letting $B = \Gamma U^\top Z N^{-1}P_0^{-1}$, we get $Z = UB$. For a scalar $\alpha > 0$, denoting $B(\alpha) = \alpha B$ and $Z(\alpha) = \alpha Z$, we further get $Z(\alpha) = UB(\alpha)$. Applying softmax function on the rows of $Z(\alpha)$, we get the probability that node $i$ belongs to community $k$

$$
\Psi_{i,k} = \begin{cases} e^\alpha/(e^\alpha + K - 1), & \text{if } \sigma_i = k, \\ 1/(e^\alpha + K - 1), & \text{otherwise.} \end{cases}
$$

When $\alpha \to \infty$, we have $\Psi_{i,k} \to \mathbb{1}_{\{\sigma_i = k\}}$. In other words, given the $K$ eigenvectors $U$ of $P$ as design matrix, and with regression coefficients $B(\alpha)$, one can recover the true community labels exactly by multinomial regression, as $\alpha \to \infty$.

In reality, we are given $\widehat{Q}$ from GNN instead of $U$. $\widehat{Q}$ and $U$ have the following relationship:

- $\text{dist}(\text{col}(\widehat{Q}_T), \text{col}(V_1)) \leq n^{-s}$ (by Theorem 1).
- $\|V_1 - U\|_2$ is controlled by the Davis-Kahan theorem.

Let $Q = Q_T$ be the output of orthogonal iteration after $T$ iterations, as defined in Lemma 8. We have $Q = V_1 W_1 + V_2 W_2$, where $\|W_2\|_2 \leq n^{-s}/2$. So

$$
Z = UB = (U - V_1)B + (Q - \widehat{Q})W_1^{-1}B + \widehat{Q}W_1^{-1}B - V_2 W_2 W_1^{-1}B,
$$

or equivalently,

$$\widehat{Q}W_1^{-1}B = Z - E,$$

where $E = \left[(U - V_1) + (Q - \widehat{Q})W_1^{-1} - V_2W_2W_1^{-1}\right]B$. Multiplying by $\alpha$, we get $\widehat{Q}W_1^{-1}B(\alpha) = \alpha(Z - E)$. Regarding $\widehat{Q}$ as design matrix, and $W_1^{-1}B(\alpha)$ as regression coefficients, the multinomial regression generates the probability that node $i$ belongs to community $k$

$$\widetilde{\Psi}_{i,k} = \frac{\exp\left\{\alpha(Z_{i,k} - E_{i,k})\right\}}{\sum_{j=1}^{K} \exp\left\{\alpha(Z_{i,j} - E_{i,j})\right\}}. \tag{44}$$

The estimated label assignment is

$$\tilde{\sigma}_i = \underset{1 \le k \le K}{\arg\max} \widetilde{\Psi}_{i,k} = \underset{1 \le k \le K}{\arg\max}(Z_{i,k} - E_{i,k}). \tag{45}$$

Note that $Z_{i\cdot}$ (the $i$th row of $Z$) is $e_{\sigma_i}^\top$, where $e_{\sigma_i} \in \mathbb{R}^K$ is the elementary vector with 1 in its $\sigma_i$-th entry, and 0 elsewhere. Then $\arg\max_{1 \le k \le K}(Z_{i,k} - E_{i,k})$ is still $\sigma_i$ if $\|E_{i\cdot}\|_{\max} < 1/2$. Define $S = \{1 \le i \le n : \|E_{i\cdot}\|_{\max} \ge 1/2\}$. Then we have $\ell_0(\sigma, \tilde{\sigma}) \le |S|/n$. On the other hand, $|S| \le \sum_{i \in S} 4\|E_{i\cdot}\|^2 \le 4\|E\|_F^2$. Therefore,

$$\ell_0(\sigma, \tilde{\sigma}) \le \frac{4}{n}\|E\|_F^2.$$

To derive an upper bound of $\|E\|_F$, first we know that the Davis-Kahan Theorem (Davis & Kahan, 1970; Yu et al., 2015) and Lemma 16 guarantees that

$$\|U - V_1\|_F \le \frac{\|A - P\|_2}{|\gamma_K|} \le \frac{O(\sqrt{np + \log n})}{|\gamma_K|}.$$

with probability at least $1 - n^{-c_0}$.

The term $(Q - \widehat{Q})W_1^{-1}$ depends on $Q - \widehat{Q}$, which is controlled well by GNN. In particular, using (42), we can get

$$\|\widehat{Q} - Q\|_F = \sqrt{\sum_{k=1}^{K} \|q_{k,T} - q_{k,T}\|^2} \le \frac{1}{2}n^{-s}.$$

Then, from $\|W_1\|_2 = 1$, we have

$$\|(Q - \widehat{Q})W_1^{-1}\|_F \le \|Q - \widehat{Q}\|_F \cdot \|W_1^{-1}\|_2 \le \frac{1}{2}n^{-s} \le \frac{O(\sqrt{np + \log n})}{|\gamma_K|}$$

when $s$ is large enough. The term $V_2W_2W_1^{-1}$ is controlled well by orthogonal iteration. Lemma 8 leads to

$$\|V_2W_2W_1^{-1}\|_F \le \|V_2\|_F \cdot \|W_2\|_2 \cdot \|W_1^{-1}\|_2 \le \sqrt{n - K} \cdot \frac{1}{2}n^{-s} \le \frac{1}{2}n^{-(s-1)} \le \frac{O(\sqrt{np + \log n})}{|\gamma_K|}$$

when $s$ is large enough.

To control for $B$, we have $\|\Gamma\|_2 = |\gamma_1|$, $\|U^\top\|_2 = 1$, and $\|ZN^{-1}\|_2 = 1/\sqrt{n_{\min}}$ by noting $(ZN^{-1})^\top ZN^{-1} = N^{-1}$. Also, since $P_0$ is a full-rank matrix with two eigenvalues $p + (K - 1)q$ and $(p - q)$, we know that $\|P_0^{-1}\|_2 = 1/(p - q)$. Therefore,

$$\|B\|_2 \le \|\Gamma\|_2 \cdot \|U^\top\|_2 \cdot \|ZN^{-1}\|_2 \cdot \|P_0^{-1}\|_2 \le \frac{|\gamma_1|}{\sqrt{n_{\min}}(p - q)}.$$

Finally, we reach the bound

$$\|E\|_F \le \left(\|U - V_1\|_F + \|(Q - \widehat{Q})W_1^{-1}\|_F + \|V_2W_2W_1^{-1}\|_F\right)\|B\|_2$$

$$\le \frac{O(\sqrt{np + \log n})}{|\gamma_K|} \cdot \frac{|\gamma_1|}{\sqrt{n_{\min}}(p - q)}.$$

The misclassification rate is bounded by

$$\ell_0(\sigma, \tilde{\sigma}) \leq \frac{O(np + \log n)}{n\gamma_K^2} \cdot \frac{\gamma_1^2}{n_{\min}(p-q)^2}.$$

From Lemma 14, we get

$$\frac{|\gamma_1|}{|\gamma_K|} = \frac{\|G\|_2}{\sigma_{\min}(G)} \leq \frac{n_{\max}(p + (K-1)q)}{n_{\min}(p-q)} \leq \frac{\beta(p + (K-1)q)}{p-q}.$$

So we have

$$\ell_0(\sigma, \tilde{\sigma}) \leq \frac{O(np + \log n)(p + (K-1)q)^2}{n^2(p-q)^4}.$$

## F  PROOF OF THEOREM 3

It is clear that $y_{i,k} = (AZ(\sigma^{(0)}))_{i,k}$ is the number of edges that node $i$ has with all nodes that are labeled as $k$ by $\sigma^{(0)}$, and $J_n Z(\sigma^{(0)})$ has identical rows where each row represents community sizes determined by $\sigma^{(0)}$. Let $n_k^{(0)} = (J_n Z(\sigma^{(0)}))_{i,k}$. Then, $q_{i,k} = (AZ(\sigma^{(0)}))_{i,k}/n^{(0)}$ is the proportion of connections of node $i$ to community $k$, and the local refinement procedure updates according to

$$\sigma_i^{(1)} = \arg \max_{k \in [K]} q_{i,k}.$$

Let $y_k = [y_{1,k}, \ldots, y_{n,k}]^\top$, $q_k = [q_{1,k}, \ldots, q_{n,k}]^\top$ for $k \in [K]$. We design the GNN illustrated in Figure 3 to approximate $q_k$.

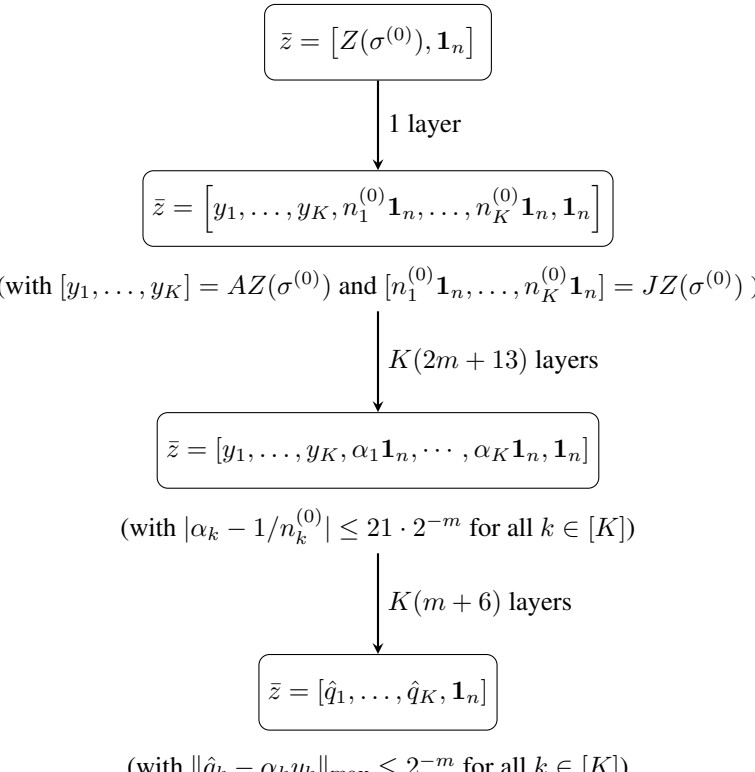

Figure 3: The GNN architecture to approximate the local refinement procedure.

The probability matrix $\widehat{\Psi}$ is given by

$$\widehat{\Psi}_{i,k} = \frac{\exp\left(\alpha \hat{q}_{i,k}\right)}{\sum_{j=1}^K \exp\left(\alpha \hat{q}_{i,k}\right)}. \tag{46}$$

for a scalar $\alpha > 0$. The estimated label is

$$\hat{\sigma}_i = \arg\max_{1 \leq k \leq K} \widehat{\Psi}_{i,k} = \arg\max_{1 \leq k \leq K} \hat{q}_{i,k}. \tag{47}$$

Since $y_{i,k} \leq n$, we have

$$\|\hat{q}_{i,k} - q_{i,k}\| \leq |\hat{q}_{i,k} - \alpha_k y_{i,k}| + |\alpha_k y_{i,k} - \frac{1}{n_k^{(0)}} y_{i,k}|$$

$$\leq 2^{-m} + 21 \cdot 2^{-m} n$$

$$\leq 22 \cdot 2^{-m} n.$$

Gao & Ma (2021) provide an argument to show that minimax bound is closely related to a fundamental hypothesis testing problem. The problem is stated as follows. Suppose we observe $X = (X_1, \ldots, X_{m_1+m_2}) \in \{0,1\}^{m_1+m_2}$, we want to test

$$H_1 : X \sim \bigotimes_{i=1}^{m_1} \text{Bernoulli}(p) \otimes \bigotimes_{i=m_1+1}^{m_1+m_2} \text{Bernoulli}(q)$$

$$\text{v.s.} \quad H_2 : X \sim \bigotimes_{i=1}^{m_1} \text{Bernoulli}(q) \otimes \bigotimes_{i=m_1+1}^{m_1+m_2} \text{Bernoulli}(p). \tag{48}$$

The local refinement procedure is designed to solve this testing problem, hence the misclassification rate after the local refinement procedure is determined by the error bound of the local refinement. Lemma 17 in Gao et al. (2017) gives detailed calculation of this error bound.

To fix ideas, we focus on the local refinement of node 1 and assume without loss of generality $\sigma_1 = 1$. One can show, by following the steps of Lemma 17 in Gao et al. (2017), that

$$\mathbb{P}(q_{1,1} \leq \max_{k \neq 1} q_{1,k}) \leq \exp\left\{-(1 + o(1)) \min_{k \neq 1} \left(\frac{n_1 + n_k}{2}\right) I(p, q)\right\}. \tag{49}$$

When $k = 2$, $\min_{k \neq 1}\left(\frac{n_1+n_k}{2}\right) = n/2$, and when $k \geq 3$, $\min_{k \neq 1}\left(\frac{n_1+n_k}{2}\right) \geq n/(\beta K)$. When using GNN to approximate the local refinement process, one needs some buffer for the difference $q_{1,1} - q_{1,k}$. Note that (49) is proved by a Chernoff bound derivation, the first step of which is

$$\mathbb{P}(q_{1,1} \leq q_{1,k}) = \mathbb{P}(e^{t^*(q_{1,k}-q_{1,1})} \geq 1) \leq \mathbb{E}(e^{t^*(q_{1,k}-q_{1,1})})$$

with $e^{t^*} = \sqrt{p(1-q)}/\sqrt{q(1-p)}$. Similarly, for any $\delta > 0$, we have

$$\mathbb{P}(q_{1,1} \leq q_{1,k} + \delta) = \mathbb{P}(e^{t^*(q_{1,k}-q_{1,1})+t^*\delta} \geq 1) \leq e^{t^*\delta}\mathbb{E}(e^{t^*(q_{1,k}-q_{1,1})}).$$

We can use the same derivation to obtain

$$\mathbb{P}(q_{1,1} \leq \max_{k \neq 1} q_{1,k} + \delta) \leq \exp\left\{-(1 + o(1))\tilde{n}I(p, q) + t^*\delta\right\}.$$

For a given constant $\epsilon > 0$, we choose $\delta$ such that $t^*\delta = \frac{\epsilon}{2}\tilde{n}I(p, q)$. Then we have

$$\mathbb{P}(q_{1,1} \leq \max_{k \neq 1} q_{1,k} + \delta) \leq \exp\left\{-(1 + o(1) - \epsilon/2)\tilde{n}I(p, q)\right\}$$

$$\leq \exp\left\{-(1 - \epsilon)\tilde{n}I(p, q)\right\}.$$

For the GNN, we choose $m$ such that

$$m = \left\lceil \frac{-\log\epsilon - \log I(p, q) + \sqrt{p(1-q)}/\sqrt{q(1-p)} + \log 88}{\log 2} \right\rceil$$

Then we can guarantee $22 \cdot 2^{-m} n \leq \delta/2$. In other words, $|\hat{q}_{i,k} - q_{i,k}| \leq \delta/2$ for all $i \in [n], k \in [K]$. Therefore,

$$\mathbb{P}(\hat{q}_{1,1} \leq \max_{k \neq 1} \hat{q}_{1,k}) \leq \mathbb{P}(q_{1,1} \leq \max_{k \neq 1} q_{1,k} + \delta)$$

$$\leq \exp\left\{-(1 - \epsilon)\tilde{n}I(p, q)\right\}.$$

We get the desired rate. The derivation from the local refinement error bound to the overall misclassification rate bound is provided in Gao et al. (2017) and is omitted here for brevity.

The depth $M''$ of the constructed GNN is $3Km + 19K + 1$, and is bounded by $3Km + 20K$.

## G  PROOFS OF THEOREMS 4 AND 5

We first define a "truncated version" of cross-entropy loss. For $L > 0$, define

$$\ell_1^{(L)}(\sigma, \Psi) = \min_{\pi \in \mathcal{S}_K} \frac{1}{n} \sum_{i \in [n]} \left\{ \left[ -\log \left( \Psi_{i, \pi(\sigma_i)} \right) \right] \wedge L \right\}.$$

for a probability matrix $\Psi \in \mathbb{R}^{n \times K}$. When $L \to \infty$, $\ell_1^{(L)}(\sigma, \Psi) \to \ell_1(\sigma, \Psi)$.

**Proof of Theorem 4.** In the proof of Theorem 2, we have constructed a GNN that, for any graph $G$ in the training set with adjacency matrix $A$, produces the probability matrix $\widetilde{\Psi}$ and estimated label assignment $\tilde{\sigma}$, defined by (44) and (45) respectively.

Note that $\tilde{\sigma}$ does not depend on $\alpha$ but $\widetilde{\Psi}$ does. Define $S' = \{1 \le i \le n : \|E_{i\cdot}\|_{\max} \ge 1/3\}$. Then for any $i \notin S'$, $Z_{i,\sigma_i} - E_{i,\sigma_i} \ge 2/3$ and $Z_{i,k} - E_{i,k} \le 1/3$ for all $k \ne \sigma_i$. We choose $\alpha$ large enough such that

$$\log(1 + (K-1)e^{-\alpha/3}) \le R.$$

Then for any $i \notin S'$, we get

$$-\log(\widetilde{\Psi}_{i,\sigma_i}) = \log \left( 1 + \sum_{j \ne \sigma_i} \exp \left\{ \alpha(Z_{i,j} - E_{i,j}) - \alpha(Z_{i,\sigma_i} - E_{i,\sigma_i}) \right\} \right)$$

$$\le \log(1 + (K-1)e^{-\alpha/3})$$

$$\le R.$$

For any $i \in S'$, we have $-\log(\widetilde{\Psi}_{i,\sigma_i}) \le L$. Then,

$$\ell_1^{(L)}(\sigma, \widetilde{\Psi}) \le R + L \frac{|S'|}{n}.$$

On the other hand, $|S'| \le \sum_{i \in S} 9\|E_{i\cdot}\|^2 \le 9\|E\|_F^2$. Using the upper bound of $\|E\|_F$ derived in the proof of Theorem 2, we get for any $c_0' > 0$,

$$\ell_1^{(L)}(\sigma, \widetilde{\Psi}) \le R + L \cdot O(R)$$

with probability at least $1 - n^{-c_0'}$. Then

$$\mathbb{E}\ell_1^{(L)}(\sigma, \widetilde{\Psi}) \le R + L \cdot O(R) + Ln^{-c_0'}.$$

We need to following lemma to proceed.

**Lemma 19.** *For any community detection algorithm $f$, suppose graphs $G_i$ are generated i.i.d. following some prior $\pi_G$, we have*

$$\mathbb{P}\left( |R(g) - \hat{R}_m(g)| \ge t \right) \le 2\exp\left( -\frac{2mt^2}{L^2} \right). \tag{50}$$

*Proof.* Note by definitions $R(g) - \hat{R}_m(g) = \frac{1}{n} \sum_{j=1}^m \ell_{g,L}(G_j) - \mathbb{E}_{G \sim \pi} \ell_{g,L}(G)$. Apply Hoeffding's inequality and use the fact that each $\ell_{g,L}(G)$ is naturally bounded by $L$. □

Suppose graph $G_i$ in the training set has true community labels $\sigma^{(i)}$, and when the GNN designed in Theorem 2 is applied to $G_i$, probability matrix $\widetilde{\Psi}^{(i)}$ is produced. Take $t = R$ in Lemma 19. We get

$$\sum_{i=1}^m \ell_1^{(L)}(\sigma^{(i)}, \widetilde{\Psi}^{(i)}) \le \mathbb{E}\ell_1^{(L)}(\sigma, \widetilde{\Psi}) + R \le 2R + L \cdot O(R) + Ln^{-c_0'}.$$

with probability at least $1 - 2\exp\left(-2mt^2/L^2\right)$.

We choose $L = R^{-\epsilon/2}$, which tends to $\infty$. Then $2R + L \cdot O(R) + Ln^{-c_0'} = O(R^{1-\epsilon/2})$ when $c_0'$ is large enough. By the continuity of $\ell_1^{(L)}$ with respect to $L$, we get

$$\sum_{i=1}^{m} \ell_1(\sigma^{(i)}, \widetilde{\Psi}^{(i)}) = O(R^{1-\epsilon/2}).$$

With $m \geq R^{-(1+\epsilon)}(\log n)^{1+\epsilon}$, one can show that the probability $1 - 2\exp\left(-2mt^2/L^2\right)$ is bigger than $1 - n^{-c_0}$ for any $c_0 > 0$.

For graph $G_i$, denote the probability matrix generated by the trained GNN as $\widetilde{\Psi}'^{(i)}$. Assuming the empirical risk is decreased, we have

$$\frac{1}{m}\sum_{i=1}^{m}\mathbb{E}\ell_1(\sigma^{(i)}, \widetilde{\Psi}'^{(i)}) \leq \frac{1}{m}\sum_{i=1}^{m}\ell_1(\sigma^{(i)}, \widetilde{\Psi}^{(i)}) \leq O(R^{1-\epsilon/2}).$$

We finally get

$$\frac{1}{m}\sum_{i=1}^{m}\ell_0(\sigma^{(i)}, \sigma(A^{(i)}, Q_0; \tilde{\theta})) \leq \frac{1}{m}\sum_{i=1}^{m}\ell_1(\sigma^{(i)}, \widetilde{\Psi}'^{(i)})/\log 2 \leq R^{1-\epsilon}.$$

**Proof of Theorem 5.** Fix any graph $G$ in the training set with adjacency matrix $A$. In the proof of Theorem 3, we have constructed a GNN that produces probability matrix $\widehat{\Psi}$ and estimated labels $\hat{\sigma}$ based on $\hat{q}_{i,k}$ by (46) and (47). To derive a bound for the cross-entropy loss, we need a better control on the difference $\hat{q}_{i,\sigma_i} - \hat{q}_{i,k}$ for $k \neq \sigma_i$. We still focus on node 1. Using the same argument as the proof of Theorem 3, one can get

$$\mathbb{P}(\hat{q}_{1,1} \leq \max_{k\neq 1}\hat{q}_{1,k} + \delta) \leq \mathbb{P}(q_{1,1} \leq \max_{k\neq 1}q_{1,k} + 2\delta)$$
$$\leq \exp\left\{-(1-2\epsilon)\tilde{n}I(p,q)\right\}.$$

Denote the right-hand-side of the last display by $R_2$. We choose $\alpha$ such that

$$\log(1 + (K-1)e^{-\alpha\delta}) \leq R_2.$$

So when $\hat{q}_{1,1} > \max_{k\neq 1}\hat{q}_{1,k} + \delta$, the cross-entropy is upper bounded by $R_2$.

Then

$$\mathbb{E}\left(\left[-\log\left(\widehat{\Psi}_{1,1}\right)\right] \wedge L\right) \leq L\mathbb{P}(\hat{q}_{1,1} \leq \max_{k\neq 1}\hat{q}_{1,k} + \delta) + R_2\left(1 - \mathbb{P}(\hat{q}_{1,1} \leq \max_{k\neq 1}\hat{q}_{1,k} + \delta)\right)$$
$$\leq LR_2 + R_2.$$

Therefore, we have

$$\mathbb{E}\ell_1^{(L)}(\sigma, \widehat{\Psi}) \leq \frac{1}{n}\sum_{i\in[n]}\mathbb{E}\left\{\left[-\log\left(\widehat{\Psi}_{i,\sigma_i}\right)\right] \wedge L\right\} \leq LR_2 + R_2.$$

Suppose graph $G_i$ in the training set has true community labels $\sigma^{(i)}$, and when the GNN designed in Theorem 3 is applied to $G_i$, probability matrix $\widehat{\Psi}^{(i)}$ is produced. Take $t = R_2$ in Lemma 19. We get

$$\sum_{i=1}^{m}\ell_1^{(L)}(\sigma^{(i)}, \widehat{\Psi}^{(i)}) \leq \mathbb{E}\ell_1^{(L)}(\sigma, \widehat{\Psi}) + R_2 \leq LR_2 + 2R_2.$$

with probability at least $1 - 2\exp\left(-2mt^2/L^2\right)$.

Choose $L = \exp\left\{(\epsilon/2)\tilde{n}I(p,q)\right\}$ which goes to $\infty$ slowly. Then $LR_2 + 2R_2 \leq \exp\left\{-(1-11\epsilon/4)\tilde{n}I(p,q)\right\}$. By the continuity of $\ell_1^{(L)}$ with respect to $L$, we have

$$\sum_{i=1}^{m}\ell_1(\sigma^{(i)}, \widehat{\Psi}^{(i)}) \leq \exp\left\{-(1-11\epsilon/4)\tilde{n}I(p,q)\right\}.$$

With $m \geq \exp(2\tilde{n}I(p,q))(\log n)^{(}1+\epsilon))$, one can prove that the probability $1 - 2\exp\left(-2mt^2/L^2\right)$ is bigger than $1 - n^{-c_0}$ for any $c_0 > 0$.

For graph $G_i$, denote the probability matrix generated by the trained GNN as $\widehat{\Psi}'^{(i)}$. Assuming the empirical risk is decreased, we have

$$\frac{1}{m}\sum_{i=1}^{m}\mathbb{E}\ell_1(\sigma^{(i)}, \widehat{\Psi}'^{(i)}) \leq \frac{1}{m}\sum_{i=1}^{m}\ell_1(\sigma^{(i)}, \widehat{\Psi}^{(i)}) \leq \exp\left\{(1 - 11\epsilon/4)\tilde{n}I(p,q)\right\}.$$

We finally get

$$\frac{1}{m}\sum_{i=1}^{m}\ell_0(\sigma^{(i)}, \sigma(A^{(i)}, Z(\tilde{\sigma}^{(i)}); \hat{\theta})) \leq \frac{1}{m}\sum_{i=1}^{m}\ell_1(\sigma^{(i)}, \widehat{\Psi}'^{(i)})/\log 2 \leq \exp\left\{(1 - 3\epsilon)\tilde{n}I(p,q)\right\}.$$

## H   PROOF OF THEOREM 6

Assume that the graph are generated i.i.d. according to some prior $\pi_G$. Suppose we are interested in a class of community detection algorithms $\mathfrak{G} = \mathrm{softmax} \circ \mathfrak{F}$. For a community detection algorithm $f_{\theta,x^{(0)}} \in \mathfrak{F}$ outputting a matrix $f_{\theta,x^{(0)}}(G) \in \mathbb{R}^{n \times K}$, which is then used to generate a probability matrix after the softmax operation, and we write $g_{\theta,x^{(0)}} = \mathrm{softmax} \circ f_{\theta,x^{(0)}} \in \mathfrak{G}$, and $g_{\theta,x^{(0)}}(G) = \mathrm{softmax}(f_{\theta,x^{(0)}}(G))$.

To establish the generalization bound, we revise the cross entropy to a "truncated" version

$$\ell_{g,L}(G) := \mathrm{CE}_{g,L}(G) := \min_{\mu \in \mathcal{S}_K} \frac{1}{n}\sum_{i=1}^{n}\left(\log((g(G))_{i,\mu(\sigma(G)_i)})^{-1} \wedge L\right),$$

where $L$ is some large but fixed constant and $\sigma(G)_i$ extracts the community assignment of node $i$ in graph $G$. We also suppress the dependence on $x^{(0)}$ in the above notation.

Define the empirical risk for any community detection algorithm $g = \mathrm{softmax} \circ f$ by $\hat{R}_m^{(L)}(g) = \hat{R}_m(f) := \frac{1}{m}\sum_{i=1}^{m}\ell_{g,L}(G_i)$. The empirical risk minimizer is defined to be $\tilde{g}^{(L)} = \arg\min_{g \in \mathfrak{G}} \frac{1}{m}\sum_{i=1}^{m}\ell_{g,L}(G_i)$. Define the population risk for any community detection algorithm $R^{(L)}(g) = \mathbb{E}_{G' \sim \pi_G}\ell_{g,L}(G')$.

For a metric space $\mathcal{F}$ equipped with metric $d$, the covering number $N(\varepsilon, \mathcal{F}, d)$ is the smallest number of balls of radius $\varepsilon$ with respect to $d$ that can cover $\mathcal{F}$, i.e., for every $\varepsilon > 0$, there exists an $\mathcal{F}_\varepsilon$ with $|\mathcal{F}_\varepsilon| = N(\varepsilon, \mathcal{F}, d)$ such that for every $f \in \mathcal{F}$, there exists a $\tilde{f} \in \mathcal{F}_\varepsilon$ such that $d(f, \tilde{f}) \leq \varepsilon$.

**Lemma 20.** *For $\delta \leq 1$ and $d_\infty(f_1, f_2) = \max_G \sup_{\|x^{(0)}\|_{\max} \leq 1} \|f_1(G, x^{(0)}) - f_2(G, x^{(0)})\|_{\max}$,*

$$\log N\left(\delta, \mathcal{G}(M, \boldsymbol{d}, \mathfrak{s}), d_\infty\right) \leq (s+1)\log\left(2\delta^{-1}n^{M+1}|\mathcal{F}|^M(M+1)V^2\right).$$

*Proof of Lemma 20.* Write $A_k^{\mathrm{L}}(f)$ as the mapping $u \in \mathbb{R}^{d_k} \mapsto \sigma_{\theta^{(M)}} \circ \cdots \circ \sigma_{\theta^{(k)}}(u) \in \mathbb{R}^{d_M}$, and $A_k^{\mathrm{R}}(f)$ as the mapping $u \in \mathbb{R}^{d_0} \mapsto \sigma_{\theta^{(k)}} \circ \cdots \sigma_{\theta^{(0)}}(u)$.

We note $\|AB\|_{\max} \leq \|A\|_{\max}\|B\|_{\max} \times$ (# of columns of $A$). Given two community detection algorithms $f$ an $\tilde{f}$ with corresponding parameters $(\theta^{(t)})_{t=0}^{M}$ and $(\tilde{\theta}^{(t)})_{t=0}^{M}$ with $\|\theta - \tilde{\theta}\|_{\max} \leq \varepsilon$, we bound the difference

$$\sup_{\|x^{(0)}\|_{\max} \leq 1} f(G, x^{(0)}) - \tilde{f}(G, x^{(0)})\|_{\max}$$

$$\leq \sum_{\ell=0}^{M}\left|A_{\ell+1}^{\mathrm{L}}(f) \circ \sigma_{\theta^{(\ell)}} \circ A_{\ell-1}^{\mathrm{R}}(\tilde{f})(X) - A_{\ell+1}^{\mathrm{L}}(f) \circ \sigma_{\tilde{\theta}^{(\ell)}} \circ A_{\ell-1}^{\mathrm{R}}(\tilde{f})(X)\right|$$

$$\leq \varepsilon(n|\mathcal{F}|)^{M+1}(M+1)\prod_{\ell=0}^{M}(d_\ell + 1).$$

where in the above we have used $\|X\|_{\max} \leq 1$, which is a simple consequence if we assume every column of $X$ has unit length. Furthermore, we have used the fact $\|\theta\|_{\max} \leq 1$ for all $\theta \in \mathcal{G}(M, \boldsymbol{d}, \mathfrak{s}, \boldsymbol{X})$, as well as that $\|O\|_{\max} \leq 1$ for any $O \in \mathcal{F}$.

The total number of parameters in $\mathcal{G}(M, \boldsymbol{b})$ is

$$U = \frac{|\mathcal{F}|}{2} \sum_{\ell=0}^{M} d_\ell d_{\ell+l} \leq \frac{|\mathcal{F}|}{2} 2^{-(M-1)}(M+1) \prod_{\ell=0}^{M}(d_\ell + 1) \leq |\mathcal{F}| \prod_{\ell=0}^{M}(d_\ell + 1) =: V.$$

We take the grid size $\delta/(n^{M+1}(M+1)|\mathcal{F}|^M V$ to discretize the active parameters on $[0,1]$, and there are $\binom{U}{\mathfrak{s}} \leq V^{\mathfrak{s}}$ ways to choose the active parameters, and therefore

$$N(\delta, \mathcal{G}(M, \boldsymbol{d}, \mathfrak{s}), d_\infty) \leq \sum_{u=1}^{\mathfrak{s}}(2\delta^{-1}n^{M+1}|\mathcal{F}|^M(M+1)V^2)^u \leq (2\delta^{-1}n^{M+1}|\mathcal{F}|^M(M+1)V^2)^{\mathfrak{s}+1}.$$

where we used the sum of the geometric sequence in the last inequality. $\qquad\square$

For the generalization bounds, we note the following facts.

**Remark 6.** *For two vectors $a \in \mathbb{R}^{1 \times k}$ and $b \in \mathbb{R}^{1 \times k}$ such that $\|a - b\|_{\max} \leq \varepsilon \leq 1$, we write $p_a = \mathrm{softmax}(a)$ and $p_b = \mathrm{softmax}(b)$. Recall $p_a = \left( \frac{\exp(a_\ell)}{\sum_{u \in [k]} \exp(a_u)} \right)_{\ell \in [k]}$. By elementary algebra, we have $\|p_a - p_b\|_{\max} \leq e^{2\varepsilon} - 1 \leq 2e^2 \varepsilon$.*

**Remark 7.** *Note that with $\|P_1 - P_2\|_{\max} \leq \delta$ for two probability matrices $P_i = (p_{k \in [n], \ell \in [K]}^{(i)} \in [0,1]^{n \times K}$, we bound for any permutation $\mu \in \mathcal{S}^K$*

$$\left| \frac{1}{n} \sum_{i \in [n]} \left( \log(p_{i, \mu(\sigma_i)}^{(1)} \vee e^{-L}) - \frac{1}{n} \sum_{i \in [n]} \log(p_{i, \mu(\sigma_i)}^{(2)}) \vee e^{-L} \right) \right| \leq \log(1 + e^L \delta) \leq e^L \delta.$$

*By Lipschitz continuity of the $\min$ functional, we have $|\mathrm{CE}(P_1) - \mathrm{CE}(P_2)| \leq e^L \delta$.*

With slight abuse of notation, we write $d_\infty(g_1, g_2) = \max_G \sup_{\|x^{(0)}\|_{\max} \leq 1} |\mathrm{CE}_{g_1}(G) - \mathrm{CE}_{g_2}(G)|$. Combining the above two remarks, we have

$$\log N\big(\delta, \mathcal{SG}(M, \boldsymbol{d}, \mathfrak{s}), d_\infty\big) \leq (\mathfrak{s}+1) \log\big(4e^{2+L}\delta^{-1}n^{M+1}|\mathcal{F}|^M(M+1)V^2\big)$$

We choose an $\delta$-covering $\mathfrak{G}_\delta$ of $\mathfrak{G} = \mathcal{SG}(M, \boldsymbol{d}, \mathfrak{s})$. By taking the standard empirical process argument, we have

$$\sup_{g \in \mathfrak{G}} |R(g) - \hat{R}_m(g)| \leq \max_{g \in \mathfrak{G}_\delta} |R(g) - \hat{R}_m(g)| + 2\delta,$$

Take union bound for all $f$'s in $\mathfrak{F}_\delta$ and apply (50), we have with probability 1 - $u$,

$$\max_{g \in \mathfrak{G}_\delta} |R(g) - \hat{R}_m(g)| \leq \sqrt{\frac{L^2}{2m}\big(\log(|\mathfrak{G}_\delta|) + \log(2/u)\big)}$$

Combining the above display with Lemma 20, we have established the desired results by taking $u = 2/m^2$ and

$$\delta = \log(L) \sqrt{\frac{2s(4+L) + 2\log(m) + (M+2)\log(n|\mathcal{F}|d_\star^2)}{2m}},$$

where $d_\star = \|\boldsymbol{d}\|_{\max}$ and note $V \leq |\mathcal{F}|d_\star^{M+1}$.

Combining the above, we have the following proposition.

**Proposition 21.** *For $\mathfrak{G} = \mathcal{SG}(M, \boldsymbol{d}, \mathfrak{s})$ and any sufficiently large but fixed $L$, with probability $1 - 2/m^2$,*

$$\sup_{g \in \mathcal{GS}(M, b, s, \boldsymbol{X})} |R^{(L)}(g) - \hat{R}_m^{(L)}(g)| \leq 3L \sqrt{\frac{2s(4+L) + 2\log m + (M+2)\log(n|\mathcal{F}|d_\star^2)}{2m}}.$$

Take $M = O(\log^2(n))$, $d_\star = O(n)$, $\mathfrak{s} = O(n \log(n))$, and $L = O(\log(n))$, the right hand side is of $O(n \log^4(n)/m)$.

Combining the above proposition and Theorem 5, we have shown Theorem 6

Assume the graphs $A$'s are generated i.i.d. following $\pi_A = \mathrm{SBM}(\boldsymbol{n}, p, q)$. Under the condition of Theorem 4, by taking $m = O\big(R^{-(1+\varepsilon)} \max((\log(n)^{1+\varepsilon}), n \log^4(n))\big)$, we have with probability $1 - n^{-c}$ for some $c < 1$, the expected mis-classification ratio on $A \sim \mathrm{SBM}(\boldsymbol{n}, p, q)$ of the trained GNN on

$$\mathbb{E}[\ell_0(\sigma, \sigma(A, Q; \tilde{\theta}) \mid \tilde{\theta})] \leq c' R^{1-\varepsilon},$$

where the constant $c'$ depends on $\varepsilon$ and $c$.

# I DETAILED CONFIGURATIONS OF NUMERICAL EXPERIMENTS AND ADDITIONAL NUMERICAL RESULTS

## I.1 SYNTHETIC EXPERIMENTS ON SBM

For SBM training set, we choose 15 logarithmically spaced values of SNR in $[0.5, 3]$, and 15 logarithmically spaced values of $C$ in $[3K, 9K]$. The community size vector $\mathbf{n}$ is determined by $n \sim \mathrm{Uniform}[500, 1500]$, multiplied by a Dirichlet-distributed random variable with parameter $\alpha \mathbf{1}_K$, where $\alpha \in \{0.3, 1.2, 3, 4, 5\}$. For each distinct combination of $(\mathrm{SNR}, C, \alpha)$, we generate 4 independent graph instances. All parameter combinations considered, the resulting training set comprises 4,500 graph instances.

The test set is constructed using combinations of SNR values from $\{0.25, 0.5, 0.75, 1, 1.5\}$ and $C$ values from $\{5, 10, 15\}$ for $K = 2$, $\{15, 20, 25\}$ for $K = 4$, and $\{25, 30, 35\}$ for $K = 8$. For each value of $K$, we define four prototypical class-size vectors, $\{\mathbf{n}^{(1)}, \mathbf{n}^{(2)}, \mathbf{n}^{(3)}, \mathbf{n}^{(4)}\}$, representing a range from balanced to extremely imbalanced community sizes. The specific community size vectors for each $K$ are as follows:

$$
\begin{aligned}
K = 2 :& \{\mathbf{n}^{(1)}, \mathbf{n}^{(2)}, \mathbf{n}^{(3)}, \mathbf{n}^{(4)}\} = \{[500, 500]^\top, [600, 400]^\top, [700, 300]^\top, [800, 200]^\top\}, \\
K = 4 :& \{\mathbf{n}^{(1)}, \mathbf{n}^{(2)}, \mathbf{n}^{(3)}, \mathbf{n}^{(4)}\} = \{[250, 250, 250, 250]^\top, [300, 250, 250, 200]^\top, \\
& \quad [400, 300, 200, 100]^\top, [700, 100, 100, 100]^\top\} \\
K = 8 :& \{\mathbf{n}^{(1)}, \mathbf{n}^{(2)}, \mathbf{n}^{(3)}, \mathbf{n}^{(4)}\} = \{[125, 125, 125, 125, 125, 125, 125, 125]^\top, \\
& \quad [150, 125, 125, 125, 125, 125, 125, 100]^\top, [200, 180, 160, 140, 120, 100, 80, 20]^\top, \\
& \quad [650, 50, 50, 50, 50, 50, 50, 50]^\top\}.
\end{aligned}
$$

For each distinct combination of $(\mathrm{SNR}, C, \mathbf{n})$, we generate 30 independent graph instances from $\mathrm{SBM}(\mathbf{n}, p, q)$, yielding 1,800 graphs in the test set.

Our two-stage GNN is configured as follows: The first-period GNN has 30 layers, 16 features, and $h = 1$. The second-period GNNs are built with 3 layers, 8 features, and $h = 0$.

Table 4 presents the complete performance of the base and two-stage GNNs, with results grouped by SNR and community size vector $\mathbf{n}$.

**Takeaways.** Overall, we observe that accuracy increases monotonically with SNR, and quickly saturates for smaller $K$. The two-stage GNN consistently delivers substantial gains in more challenging regimes, specifically with larger community counts ($K=8$) and low-to-moderate SNR. For instance, at $K = 8$ and $\mathrm{SNR} = 0.75$, the two-stage model improves accuracy from 77.1% to 82.7% for balanced communities ($\mathbf{n}^{(1)}$), and from 87.8% to 94.7% for moderately imbalanced communities ($\mathbf{n}^{(3)}$). This improvement is also accompanied by a reduction in variance, as seen in the $\mathbf{n}^{(3)}$ case where the standard deviation decreases from 1.05 to 0.66.

In easier regimes, such as with small $K$ and high SNR ($\geq 0.75$), improvements are negligible due to ceiling effects. We also observe a slight performance decrease in a few cases under extreme class

Table 4: Test accuracy of base and two-stage GNNs on the SBM. Note: All values are percentages, reported in the mean (standard deviation) format.

| n | SNR | $K = 2$ | | $K = 4$ | | $K = 8$ | |
|---|---|---|---|---|---|---|---|
| | | Base | Two-stage | Base | Two-stage | Base | Two-stage |
| $\mathbf{n}^{(1)}$ | 0.25 | 52.8 (3.26) | 52.9 (3.34) | 47.2 (13.2) | 50.3 (16.1) | 44.1 (1.13) | 50.4 (2.16) |
| | 0.50 | 94.8 (1.91) | 95.4 (1.35) | 93.6 (1.20) | 95.5 (0.45) | 70.5 (1.10) | 77.7 (1.05) |
| | 0.75 | 98.8 (0.06) | 99.0 (0.05) | 98.9 (0.18) | 99.0 (0.08) | 77.1 (0.76) | 82.7 (0.73) |
| | 1.00 | 99.6 (0.05) | 99.7 (0.05) | 99.7 (0.11) | 99.8 (0.04) | 78.5 (1.42) | 83.5 (0.20) |
| | 1.50 | 100 (0.00) | 100 (0.02) | 99.9 (0.17) | 100 (0.01) | 81.5 (1.35) | 83.1 (1.33) |
| $\mathbf{n}^{(2)}$ | 0.25 | 73.1 (8.43) | 75.0 (7.03) | 57.7 (10.7) | 62.8 (11.6) | 47.9 (0.57) | 54.4 (1.55) |
| | 0.50 | 96.1 (0.15) | 96.4 (0.17) | 94.7 (0.05) | 96.0 (0.38) | 71.6 (0.47) | 81.3 (0.77) |
| | 0.75 | 98.9 (0.02) | 99.0 (0.05) | 99.0 (0.12) | 99.0 (0.03) | 79.0 (1.31) | 85.5 (0.55) |
| | 1.00 | 99.7 (0.03) | 99.7 (0.03) | 99.6 (0.25) | 99.8 (0.05) | 81.2 (0.81) | 87.6 (1.10) |
| | 1.50 | 100 (0.01) | 100 (0.01) | 100 (0.06) | 100 (0.01) | 81.9 (1.59) | 86.5 (0.66) |
| $\mathbf{n}^{(3)}$ | 0.25 | 81.4 (4.15) | 83.0 (3.01) | 77.4 (2.77) | 80.4 (1.54) | 68.3 (1.83) | 72.9 (1.71) |
| | 0.50 | 96.3 (0.16) | 96.6 (0.05) | 96.2 (0.14) | 96.6 (0.13) | 83.5 (0.50) | 89.7 (0.99) |
| | 0.75 | 98.9 (0.03) | 99.1 (0.05) | 98.9 (0.10) | 99.0 (0.07) | 87.8 (1.05) | 94.7 (0.66) |
| | 1.00 | 99.7 (0.01) | 99.7 (0.04) | 99.6 (0.09) | 99.7 (0.08) | 88.8 (0.33) | 95.5 (0.35) |
| | 1.50 | 100 (0.02) | 100 (0.02) | 100 (0.01) | 100 (0.01) | 90.4 (1.26) | 97.2 (0.36) |
| $\mathbf{n}^{(4)}$ | 0.25 | 86.3 (2.20) | 87.2 (1.64) | 80.2 (0.40) | 79.7 (0.65) | 51.8 (4.61) | 59.4 (7.98) |
| | 0.50 | 96.9 (0.11) | 97.2 (0.05) | 84.6 (0.20) | 85.7 (0.34) | 56.5 (5.05) | 63.5 (5.14) |
| | 0.75 | 99.1 (0.03) | 99.2 (0.04) | 89.1 (0.82) | 91.2 (0.53) | 63.8 (9.09) | 70.6 (5.81) |
| | 1.00 | 99.7 (0.02) | 99.7 (0.03) | 93.0 (0.84) | 94.4 (0.40) | 69.6 (5.95) | 76.4 (1.81) |
| | 1.50 | 100 (0.01) | 100(0.01) | 97.5 (0.38) | 97.3 (0.45) | 80.0 (4.06) | 85.7 (6.31) |

imbalance (e.g., at $K = 4$, $\mathbf{n}^{(4)}$ and $\mathrm{SNR} = 0.25$), where the base model marginally outperforms the two-stage model. This behavior is likely due to majority-class drift during the self-training phase.

## I.2 SYNTHETIC EXPERIMENTS ON MIXED

To assess our model's robustness to model mis-specification and its generalization capabilities, we conducted experiments with three distinct training datasets: SBM, a combination of SBM and DCBM, and a combination of SBM, DCBM, and LSM. The trained models were then evaluated on SBM, DCBM, and LSM test sets. The GNN architecture for this study is configured identically to the one used in our synthetic SBM experiments, as described in Appendix I.1.

**SBM+DCBM training data.** For a fixed $K$ and each $(C, \mathrm{SNR}, \alpha)$, we draw one graph from SBM or DCBM with equal probability. We set $p = (a \log n)/n$ and $q = (b \log n)/n$, where $(a, b)$ are uniquely determined by $C = a + (K - 1)b$ and $\mathrm{SNR} = (a - b)^2/[K(a + (K - 1)b)]$ under $a > b > 0$. We form the block matrix $B$ with $B_{rr} = p$ and $B_{rs} = q$ for $r \neq s$, and then inject structured heterogeneity as follows: apply a mild diagonal jitter $B_{rr} \leftarrow B_{rr} \cdot \exp(\xi_r)$ with $\xi_r \sim \mathrm{Unif}[-\sigma_p, \sigma_p]$ (we use $\sigma_p = 0.08$); and apply an off–diagonal multiplicative mask $B_{rs} \leftarrow B_{rs} \cdot M_{rs}$ for $r \neq s$, where $M_{rr} = 1$ and the off–diagonal entries are symmetrized and renormalized so that $\mathrm{mean}_{r<s} M_{rs} = 1$. Unless otherwise noted, we adopt a *low–rank* mask $M = \sigma(\alpha_0 + UU^\top)$ with $U \in \mathbb{R}^{K \times d}$, $d = 2$, $\alpha_0 = 0$, and logistic $\sigma(\cdot)$; an optional fixed seed can be used to control randomness. For ablations, we also consider *lognormal* masks $M = \exp(G)$ with symmetric Gaussian $G$ of s.d. $\sigma_m = 0.35$, *beta* masks with entrywise $\mathrm{Beta}(a, b)$ using $(a, b) = (2, 6)$, and a *tiered* mask where communities are partitioned into three groups (default equal sizes) and $M_{rs} = \tau_{g(r), g(s)}$ with $\tau = 1 + sR$, symmetric $R \sim \mathcal{N}(0, 0.25)$, and scale $s = 0.6$; all masks are symmetrized and renormalized as above. In the DCBM case, node factors $\{\theta_i\}$ are sampled i.i.d. from $\Gamma(\kappa, 1/\kappa)$ with $\kappa$ drawn log–uniformly from $[1.5, 3]$. For each $K$, the training set contains 4,500 graphs.

**Latent Space Model** In the LSM, the edge probability between nodes $i$ and $j$ is

$$P_{ij} = \mathrm{sigmoid}\big(b_i + b_j + \phi_i^\top \phi_j\big), \quad b_i \sim \mathcal{N}(\bar{b}, 1), \ \phi_i \sim \mathcal{N}(\boldsymbol{\mu}_{\sigma_i}, \tau^2 \boldsymbol{I}).$$

The global bias $\bar{b}$ satisfies $\exp(2\bar{b}) = C \log n / n$. To construct community embeddings $\boldsymbol{\mu}(1), \ldots, \boldsymbol{\mu}(K)$, we sample $K$ vectors uniformly on the $K$-simplex and scale by $s > 0$ to control separation. This yields approximate within/between-community probabilities

$$p \approx \mathrm{sigmoid}(2\bar{b} + s^2), \qquad q \approx \mathrm{sigmoid}\Big(2\bar{b} - \frac{s^2}{K-1}\Big),$$

and we tune $s$ to match a target $\mathrm{SNR} = n\,(p-q)^2 / \big(\log n \cdot K(p + (K-1)q)\big)$.

To induce additional heterogeneity, we apply community-wise multipliers $\mathbf{r}$ to the embeddings: $\mathbf{r} = [1.0, 1.2]$ for $K = 2$, $[0.8, 0.9, 1.0, 1.1]$ for $K = 4$, and an arithmetic sequence from $0.85$ to $1.2$ for $K = 8$.

**SBM+DCBM+LSM training data**   We vary $\tau \in \{0, 0.25, 0.5\}$. When $\tau > 0$ the graphs follow the LSM; when $\tau = 0$ the model reduces to DCBM. In particular, setting $b_i \sim \mathcal{N}(\bar{b}, 1)$ yields DCBM, while $b_i \equiv \bar{b}$ recovers SBM. For each fixed $K$, we assemble a training set of $4{,}800$ graphs by sampling 10 log-spaced $\mathrm{SNR} \in [0.5, 3]$, 10 values of $C \in [3K, 9K]$, $\alpha \in \{0.3, 1.2, 3, 5\}$ for the Dirichlet size prior, and *four* independent realizations per $(\mathrm{SNR}, C, \alpha, \tau)$.

**Test data**   To comprehensively assess performance, we evaluate the models on three distinct test sets, each comprising $1{,}800$ graphs per K.

For the SBM and DCBM test sets, we maintain the same configurations of $C$, SNR, and $\mathbf{n}$ as in the first experiment. For each parameter combination $(C, \mathrm{SNR}, \mathbf{n})$, we generate 30 independent graph instances from the SBM and another 30 from the DCBM. The degree correction parameters for the DCBM are sampled in the same manner as during the training phase.

The LSM test data is constructed as follows. We generate test graphs with $\tau = 0.25$ and the same $\mathbf{r}$ multipliers as in training. For each $K$, we take $\mathrm{SNR} \in \{0.25, 0.5, 0.75, 1, 1.5\}$ and $C \in \{5, 10, 15\}$ for $K = 2$, $\{15, 20, 25\}$ for $K = 4$, and $\{25, 30, 35\}$ for $K = 8$, and the four $\mathbf{n}^{(m)}$ above. For each $(\mathrm{SNR}, C, \mathbf{n})$ we draw 30 i.i.d. graphs, yielding $1{,}800$ test graphs per $K$.

Next, Tables 5, 6, and 7 present detailed results for our synthetic experiments on mixed models. The results are stratified by community-size configuration and SNR. The columns labeled 'DCBM' and 'LSM' represent models trained on SBM+DCBM and SBM+DCBM+LSM data, respectively. The accuracy values shown are averaged over three distinct test datasets generated from SBM, DCBM, and LSM.

**Takeaways.**   As shown in Table 5, 6 and 7, our two-stage GNN consistently outperforms the base model across all tables. This advantage becomes particularly clear as the problem complexity increases, as seen with larger community counts (K=8) and smaller SNR. While our models show a performance drop on these test sets compared to the model trained on SBM dataset, we attribute this to the training strategy. With a fixed training set size, introducing more diverse graph models (SBM+DCBM+LSM) reduces the parameter coverage for each individual model. This trade-off can limit the model's ability to perfectly capture the nuances of each graph type, resulting in lower overall test accuracy, yet our two-stage architecture still manages to extract a performance gain.

### I.3   REAL DATA EXPERIMENTS

We adopt two training settings—SBM, a mixture of SBM and DCBM —and train both first- and second-period GNNs under each.For the *first-period GNN*, we use 30 layers with 16 features and h=1 when $K \in \{2, 4\}$, and 30 layers with 32 features and h=1 when $K \in \{8, 9\}$. For the *second-period GNN*, we adopt a lighter architecture: 3 layers with 8 features and h=0 when $K \in \{2, 4\}$, and 3 layers with 16 features and h=0 when $K \in \{8, 9\}$.

We evaluate on five real-world networks: the *Political Blog* network (1,222 nodes, 16,714 edges, 2 communities; Adamic & Glance (2005)); *Simmons College* (1,137 nodes, 24,257 edges, 4 communities) and *Caltech* (590 nodes, 12,822 edges, 8 communities; Traud et al. (2011; 2012)), both preprocessed following Chen et al. (2018); a *manufacturing company* network (74 nodes, 235 edges, 4 communities; Weng & Feng (2022)); and the *French high school friendship* network (329 nodes, 5,818 edges, 9 communities; Mastrandrea et al. (2015)). Table 3 reports accuracy (%) across these datasets.

Table 5: Accuracy (%) by class sizes and SNR for $K = 2$; columns group three test models with subcolumns `Base`/ `Two-stage`.

| n | SNR | SBM | | DCBM | | LSM | |
|---|---|---|---|---|---|---|---|
| | | Base | Two-stage | Base | Two-stage | Base | Two-stage |
| $\mathbf{n}^{(1)}$ | 0.25 | 54.6 (6.8) | 55.6 (6.6) | 53.7 (5.3) | 54.1 (6.4) | 51.1 (2.0) | 50.9 (2.4) |
| | 0.5 | 87.4 (14.2) | 90.5 (11.2) | 63.2 (18.9) | 63.4 (19.8) | 53.7 (8.9) | 53.8 (10.1) |
| | 0.75 | 89.6 (18.3) | 94.3 (9.3) | 66.7 (21.4) | 66.1 (22.0) | 57.0 (12.9) | 57.1 (14.3) |
| | 1.0 | 94.2 (14.1) | 98.1 (4.6) | 72.0 (21.8) | 70.6 (22.3) | 58.6 (15.7) | 58.8 (17.0) |
| | 1.5 | 97.8 (6.4) | 99.9 (0.1) | 80.4 (22.0) | 78.6 (22.1) | 60.9 (17.0) | 61.1 (18.3) |
| $\mathbf{n}^{(2)}$ | 0.25 | 70.4 (9.9) | 72.7 (9.2) | 62.9 (5.4) | 63.7 (5.9) | 59.6 (1.3) | 60.3 (0.7) |
| | 0.5 | 89.0 (14.8) | 92.0 (10.0) | 70.5 (16.1) | 70.8 (16.2) | 61.9 (6.1) | 62.8 (6.4) |
| | 0.75 | 91.3 (16.0) | 95.4 (8.1) | 73.6 (17.6) | 74.2 (17.4) | 63.6 (9.6) | 64.4 (9.7) |
| | 1.0 | 95.5 (11.2) | 99.2 (1.5) | 78.4 (18.0) | 78.0 (18.2) | 65.5 (11.5) | 66.6 (11.8) |
| | 1.5 | 98.6 (4.1) | 100.0 (0.0) | 85.1 (16.8) | 83.3 (17.7) | 67.4 (13.4) | 68.4 (13.7) |
| $\mathbf{n}^{(3)}$ | 0.25 | 76.6 (10.8) | 78.0 (10.7) | 71.0 (3.3) | 71.8 (3.2) | 69.0 (1.9) | 70.3 (1.0) |
| | 0.5 | 89.6 (13.6) | 91.8 (9.2) | 77.1 (10.9) | 77.4 (10.9) | 70.9 (5.8) | 72.3 (5.3) |
| | 0.75 | 93.5 (11.7) | 96.8 (4.6) | 80.1 (12.2) | 80.2 (12.4) | 72.0 (6.9) | 73.2 (6.6) |
| | 1.0 | 96.1 (9.0) | 98.9 (2.2) | 83.2 (13.2) | 82.9 (13.5) | 73.1 (8.6) | 74.7 (8.3) |
| | 1.5 | 97.3 (7.8) | 99.7 (0.8) | 88.3 (12.2) | 88.1 (12.7) | 75.4 (11.0) | 76.8 (10.6) |
| $\mathbf{n}^{(4)}$ | 0.25 | 80.4 (12.6) | 80.6 (13.9) | 78.9 (1.5) | 79.7 (0.3) | 77.7 (2.7) | 79.7 (0.7) |
| | 0.5 | 89.1 (15.0) | 89.1 (15.6) | 82.9 (5.6) | 83.6 (5.4) | 78.9 (3.6) | 80.9 (2.1) |
| | 0.75 | 92.2 (14.6) | 92.8 (13.6) | 85.2 (7.3) | 85.5 (7.1) | 79.4 (3.9) | 81.2 (2.7) |
| | 1.0 | 94.3 (13.5) | 95.5 (11.8) | 87.5 (8.3) | 87.5 (8.2) | 80.3 (5.6) | 82.2 (4.4) |
| | 1.5 | 95.5 (13.0) | 96.1 (11.3) | 91.0 (8.4) | 90.9 (8.0) | 81.9 (6.4) | 83.5 (5.8) |

From the table  8,Models trained on SBM+DCBM consistently outperform those trained solely on SBM. This is because real-world networks often exhibit heterogeneity, a structure not adequately captured by SBM-only training. By incorporating DCBM, the model learns to recognize this heterogeneity, which enhances its generalization ability to real data. Furthermore, applying the second-period GNN reliably improves performance on these datasets by performing local refinement to achieve more accurate community assignments.

Table 6: Accuracy (%) by class sizes and SNR for $K = 4$; columns group three test models with subcolumns `Base`/ `Two-stage`.

| n | SNR | SBM | | DCBM | | LSM | |
|---|---|---|---|---|---|---|---|
| | | Base | Two-stage | Base | Two-stage | Base | Two-stage |
| $\mathbf{n}^{(1)}$ | 0.25 | 54.7 (9.1) | 59.6 (10.2) | 36.5 (13.7) | 36.8 (15.6) | 26.0 (0.7) | 26.0 (0.8) |
| | 0.5 | 91.8 (4.0) | 94.5 (2.1) | 56.5 (24.2) | 57.8 (25.5) | 26.7 (2.0) | 26.9 (1.7) |
| | 0.75 | 97.4 (2.3) | 98.3 (1.2) | 69.8 (24.0) | 72.4 (23.1) | 27.6 (4.0) | 27.8 (3.4) |
| | 1.0 | 98.0 (2.8) | 99.0 (1.5) | 80.0 (17.1) | 83.0 (16.9) | 29.6 (7.4) | 30.7 (7.7) |
| | 1.5 | 99.5 (0.9) | 99.9 (0.3) | 90.6 (6.0) | 93.8 (4.1) | 35.1 (13.7) | 37.2 (13.8) |
| $\mathbf{n}^{(2)}$ | 0.25 | 62.8 (7.4) | 68.1 (7.3) | 39.3 (13.2) | 40.4 (14.8) | 29.9 (0.4) | 29.8 (0.2) |
| | 0.5 | 94.1 (1.7) | 95.8 (0.7) | 59.2 (22.8) | 61.2 (23.8) | 30.2 (0.9) | 30.1 (0.3) |
| | 0.75 | 98.5 (0.7) | 99.0 (0.2) | 74.8 (21.5) | 76.6 (21.9) | 30.7 (2.3) | 30.7 (1.3) |
| | 1.0 | 99.5 (0.7) | 99.7 (0.2) | 83.7 (15.4) | 86.3 (14.6) | 32.3 (5.4) | 32.9 (5.7) |
| | 1.5 | 99.8 (0.3) | 100.0 (0.0) | 93.2 (5.2) | 95.5 (3.1) | 35.0 (9.2) | 36.4 (10.2) |
| $\mathbf{n}^{(3)}$ | 0.25 | 77.9 (2.1) | 80.2 (1.8) | 52.0 (13.4) | 53.6 (14.0) | 39.2 (1.2) | 39.0 (1.1) |
| | 0.5 | 94.9 (1.2) | 95.8 (0.6) | 71.6 (18.5) | 74.3 (17.1) | 39.3 (1.1) | 39.2 (1.2) |
| | 0.75 | 98.2 (0.6) | 98.7 (0.3) | 80.6 (15.7) | 83.0 (14.4) | 39.7 (1.5) | 39.7 (1.6) |
| | 1.0 | 99.3 (0.3) | 99.5 (0.2) | 87.3 (9.9) | 89.7 (9.3) | 40.6 (3.4) | 40.6 (3.6) |
| | 1.5 | 99.8 (0.2) | 99.9 (0.0) | 93.0 (5.3) | 95.5 (3.2) | 42.7 (6.6) | 43.8 (7.1) |
| $\mathbf{n}^{(4)}$ | 0.25 | 74.0 (9.5) | 75.8 (6.1) | 63.8 (9.0) | 65.5 (8.2) | 65.6 (5.7) | 63.3 (7.3) |
| | 0.5 | 83.1 (4.3) | 84.8 (2.0) | 69.9 (11.4) | 69.8 (13.5) | 66.2 (5.4) | 63.3 (8.3) |
| | 0.75 | 88.4 (1.9) | 90.3 (1.6) | 73.9 (12.1) | 74.5 (14.0) | 66.5 (5.1) | 62.6 (8.9) |
| | 1.0 | 92.7 (2.0) | 94.3 (1.7) | 76.8 (12.4) | 78.6 (13.1) | 67.0 (4.7) | 63.6 (9.0) |
| | 1.5 | 97.2 (1.2) | 97.7 (0.9) | 81.8 (12.5) | 85.2 (10.7) | 67.8 (4.6) | 63.8 (8.8) |

Table 7: Accuracy (%) by class sizes and SNR for $K = 8$; columns group three test models with subcolumns `Base`/ `Two-stage`.

| n | SNR | SBM | | DCBM | | LSM | |
|---|---|---|---|---|---|---|---|
| | | Base | Two-stage | Base | Two-stage | Base | Two-stage |
| $\mathbf{n}^{(1)}$ | 0.25 | 40.6 (7.8) | 47.1 (8.0) | 26.7 (8.9) | 31.9 (12.2) | 15.0 (1.2) | 14.2 (1.5) |
| | 0.5 | 65.5 (7.5) | 77.3 (2.3) | 43.9 (13.0) | 55.3 (17.1) | 15.4 (1.6) | 14.7 (2.1) |
| | 0.75 | 73.0 (4.0) | 83.6 (3.0) | 55.5 (10.3) | 68.0 (12.4) | 15.9 (2.2) | 15.3 (2.9) |
| | 1.0 | 75.0 (3.0) | 84.3 (4.8) | 62.7 (7.4) | 74.8 (9.3) | 17.4 (3.8) | 17.4 (5.1) |
| | 1.5 | 78.2 (3.0) | 85.2 (6.0) | 69.9 (5.7) | 79.3 (7.9) | 19.1 (5.7) | 20.0 (7.9) |
| $\mathbf{n}^{(2)}$ | 0.25 | 43.1 (8.8) | 49.9 (8.9) | 27.5 (8.4) | 32.4 (11.5) | 16.0 (0.6) | 15.7 (0.6) |
| | 0.5 | 66.6 (8.6) | 79.1 (3.6) | 44.6 (12.9) | 56.3 (15.5) | 16.4 (1.1) | 16.1 (1.0) |
| | 0.75 | 74.0 (5.4) | 85.2 (1.4) | 56.2 (10.7) | 69.2 (12.3) | 16.8 (1.4) | 16.6 (1.6) |
| | 1.0 | 76.6 (4.0) | 87.5 (3.5) | 64.2 (7.0) | 75.5 (8.3) | 17.6 (2.5) | 18.2 (3.4) |
| | 1.5 | 78.8 (2.5) | 87.1 (4.4) | 70.6 (6.1) | 80.3 (6.7) | 19.1 (4.1) | 19.8 (5.1) |
| $\mathbf{n}^{(3)}$ | 0.25 | 60.6 (12.3) | 66.8 (10.2) | 40.1 (9.5) | 46.5 (12.2) | 20.1 (0.2) | 20.2 (0.1) |
| | 0.5 | 79.7 (5.4) | 87.2 (3.6) | 60.4 (9.7) | 70.2 (9.9) | 20.2 (0.1) | 20.4 (0.4) |
| | 0.75 | 85.0 (2.8) | 92.6 (1.8) | 70.7 (6.4) | 78.5 (7.1) | 20.6 (0.5) | 20.9 (0.8) |
| | 1.0 | 86.3 (2.1) | 93.6 (1.7) | 75.3 (5.7) | 83.1 (6.0) | 20.9 (1.0) | 21.6 (1.6) |
| | 1.5 | 87.7 (2.2) | 95.0 (2.1) | 80.6 (3.5) | 88.3 (3.4) | 22.2 (2.2) | 23.4 (3.7) |
| $\mathbf{n}^{(4)}$ | 0.25 | 62.8 (10.0) | 61.5 (8.3) | 62.6 (4.8) | 64.7 (2.0) | 52.5 (8.2) | 58.9 (6.8) |
| | 0.5 | 69.0 (9.9) | 71.2 (6.6) | 70.5 (3.0) | 70.7 (2.9) | 53.1 (8.6) | 59.7 (5.9) |
| | 0.75 | 73.1 (8.6) | 75.7 (5.0) | 73.2 (3.4) | 73.6 (4.0) | 55.4 (6.6) | 61.0 (4.8) |
| | 1.0 | 76.4 (6.6) | 78.8 (2.3) | 75.2 (3.9) | 76.2 (5.0) | 57.9 (4.7) | 63.0 (2.6) |
| | 1.5 | 81.6 (4.0) | 83.8 (3.7) | 77.5 (4.6) | 78.8 (5.1) | 59.0 (4.0) | 64.2 (2.0) |

Table 8: Real-world evaluation: accuracy (%) across five datasets. Stage-2 applies the second period GNN.

| Dataset | SBM | | SBM+DCBM | |
|---|---|---|---|---|
| | Base | Two-stage | Base | Two-stage |
| Political Blog | 89.2% | 93.3% | 94.8% | 95.3% |
| Simmons | 73.0% | 73.7% | 73.3% | 77.5% |
| Caltech | 46.9% | 62.7% | 70.3% | 74.6% |
| Company | 94.5% | 94.5% | 94.5% | 96.0% |
| High school | 73.6% | 85.4% | 89.4% | 98.5% |

