# OpenReview forum: "Optimal community detection with graphical neural networks"
_ICLR.cc/2026/Conference — Submitted to ICLR 2026_

### Official Review · Reviewer_mpcZ · 2025-10-28

**Soundness:** 2
**Presentation:** 2
**Contribution:** 2
**Rating:** 2
**Confidence:** 4

**Summary:**

The present work treats community detection as a supervised learning task and investigates theoretical performance guarantees for GNNs applied to unsupervised community detection. The authors show that GNNs with ReLU activations can achieve information-theoretic optimality on graphs generated by a stochastic block model. They also quantify the depth GNNs require for this. Using a two-stage setup, the authors evaluate the performance of GNNs on synthetic network datasets generated from stochastic block models, and on empirical datasets. They show that the two-stage setup they consider outperforms a simpler GNN baseline model.

I believe that considering what performance guarantees can be given for GNNs is an important research question. While GNNs deliver state-of-the-art performance across many applications, it is often necessary to have a "good hunch" and/or perform extensive hyperparameter tuning to achieve reproducible, high-performance results. There are simply many moving parts, and I believe it would be useful to have theoretical performance guarantees. However, I am unsure whether the theoretical results established in this work carry over into practice. Why? If I understood the paper correctly, much of it relies on assumptions: first, that the training data is generated from a stochastic block model. But this assumes pairwise-independent links, whereas in practice, network links are often not pairwise independent. Second, that ReLu activations are used. But if I am not mistaken, the field is phasing out ReLUs in favour of other activation functions such as ELU or SELU. Third, the theorems state that "... there exists a GNN ...", but, as always, trainability is a different issue, which the authors seem to hint at in Remark 5. Moreover, having worked with community detection myself, I am a bit surprised that this work treats it as a supervised problem, since it is most widely considered a fundamentally unsupervised one. I am also a bit surprised that the authors consider the misclassification rate as a performance measure instead of, for example, adjusted mutual information, normalised mutual information, or the Jaccard index, which are more commonly used to measure community quality.

I also have some concerns regarding the synthetic data. From the paper's description, it is not clear how the node features were generated or how they were used in training (see questions below).

**Strengths:**

- The authors consider a relevant research question, namely, possible performance guarantees for GNNs, and establish theoretical results, accompanied by mathematical proofs.
- The authors perform numerical studies to validate the performance of their two-stage GNN scheme and find that it outperforms

**Weaknesses:**

- Community detection is a fundamentally unsupervised task, however, in this work, the authors diverge from the established standard and consider it as a supervised task. One reason that community detection is most widely considered unsupervised is that, in practice, it is infeasible to obtain ground-truth communities [1].
- The paper lacks a related work section that briefly reviews the literature for community detection with graph neural networks, such as [1,2,3,4,5]. I do see a hint at previous work in the introduction, but I believe that this short section does not position the present paper sufficiently well in the literature. Perhaps, also, the connection to graph pooling should be discussed, which seems to have a setup more similar to the present work.
- Given that the authors hint at learnability as a problem in Remark 5, it seems as though it remains open to what extent the theoretical results carry over into practical settings.

[1] L. Peel et al., The ground truth about metadata and community detection in networks, Science Advances, 2017. \
[2] R. Ying et al., Hierarchical Graph Representation Learning with Differentiable Pooling, NeurIPS, 2018. \
[3] O. Shchur and S. Günnemann, Overlapping Community Detection with Graph Neural Networks, Deep Learning on Graphs Workshop, KDD, 2019. \
[4] F. Bianchi et al., Spectral Clustering with Graph Neural Networks for Graph Pooling, ICML, 2020. \
[5] A. Tsitsulin et al., Graph Clustering with Graph Neural Networks, JMLR, 2023. \
[6] C. Blöcker et al, The Map Equation Goes Neural: Mapping Flows on Networks with Graph Neural Networks, NeurIPS, 2024.

**Questions:**

There are a couple of points that remained unclear to me after reading the paper. I am looking forward to the author's replies to clarify those points.

1. I found the wording in the abstract somewhat misleading on my first read: I thought that "achieving information-theoretic optimality under the stochastic block model" means that the authors consider fitting an SBM, given some data, and claim to achieve information-theoretic optimality in doing so. However, I now believe this means that, when the data is generated with an SMB, then it is possible to perform information-theoretically optimal supervised community detection with a GNN. Is that right?
2. Connected to the previous question, I am wondering how much data is required to achieve information-theoretic optimality? I assume that one sample (one graph instance) is probably not enough. Two samples? Three samples? I suppose that it depends on the network size (number of nodes) and density (number of links). Can you provide some insights?
3. The theoretical results seem to be only valid for graphs generated from a simple stochastic block model. What if the data is generated according to some other rule, such as LFR networks? Or empirical networks, which tend to be noisy, and with unknown ground truth? Do your established guarantees still apply? My intuition tells me it's hard to say without knowing the ground truth —is that right?
4. What is the reason behind using the misclassification rate as a measure of community quality? In network science, standard measures of community-detection performance include variants of mutual information, such as adjusted mutual information and normalised mutual information, and the Jaccard index, but rarely accuracy/misclassification rate.
5. In recent years, several activation functions have been introduced, including ELU and SELU. Can your results be transferred/adapted to other activation functions, or is it essential that ReLU is used?
6. It seems like the synthetic and empirical networks used in the numerical studies are relatively small. This leaves me wondering about scalability. Could you say something about how your two-stage approach scales? I am also a bit concerned about the number of layers. From a brief look at the appendix, I saw that the first stage uses 30 layers, which might make the approach difficult to use in practice.
7. How were the synthetic networks generated? Specifically, let's assume we generate several networks with 1000 nodes, K = 2 communities, and community sizes [500,500]. How are the community memberships assigned then? Will the first 500 nodes always belong to the first community? And what node features are used? I am asking because if the first 500 nodes always belong to the first community and the node features encode node IDs, then the GNN may actually learn that nodes with IDs 1-500 belong to the first community, while ignoring actual topological patterns that should be used to determine communities.
8. I am also wondering what happens when we train on small networks but want to predict communities in much larger networks? Say, the training networks have at most 10 communities. But now we predict on networks with 100 communities. How does your two-stage approach handle this inductive setting, especially when there are many more communities in the validation data? (I simply assume that the intention is to apply the trained networks to unseen networks whose number of communities is, in general, unknown.)
9. What is the connection between your work and deep graph pooling? I am asking because the question "Can the trained GNN achieve strong performance for out-of-sample networks?" sounds very similar to the inductive graph classification setting typically considered in graph pooling.
10. Do I understand it correctly that Remark 5 refers to trainability? That is, even though you can prove that there is a GNN that achieves information-theoretic optimality, you acknowledge that it may be difficult or infeasible to find it in practice?

Minor points
- I find it a bit strange that GCN and GNN are mentioned as examples for deep community detection in the introduction. GCN is a specific GNN architecture, and I would not consider it an algorithm for community detection. Instead, they are building blocks used in deep community detection, together with some community-detection objective that quantifies what good communities are.
- L.91 "c is an positive" should be "c is a positive"
- I believe that, when $G \sim SMB(n,p,q)$ is mentioned in l.261, it is not formally defined what $SBM(n,p,q)$ means.
- Duplicate "the fact the fact" in l.276
- Typo "mischassification" in l.376

---

### Official Review · Reviewer_YFoR · 2025-10-30

**Soundness:** 2
**Presentation:** 2
**Contribution:** 2
**Rating:** 2
**Confidence:** 4

**Summary:**

This paper investigates the theoretical optimality of community detection within the GNN framework. It proposes a two-stage GNN training scheme and proves that GNNs can achieve the information-theoretic limit under the Stochastic Block Model (SBM), matching the performance of classical spectral clustering and likelihood-based methods. The core contributions include: (1) establishing a statistical theoretical foundation for GNNs in community detection and deriving both approximation error bounds and minimax optimal misclassification rates; (2) introducing a two-stage GNN training strategy that provides a reusable analytical framework, in which the generalization bounds of GNN-based community detection algorithms are derived through an in-depth study of the complexity of the underlying GNN function class; and (3) conducting extensive experiments that validate the theoretical results, demonstrating strong robustness and generalization capability across various network settings.

**Strengths:**

1. The paper demonstrates that a properly designed GNN can achieve information-theoretic optimality for community detection under the SBM. This work bridges a critical theoretical gap by elevating GNNs from empirical tools to methods with rigorous statistical guarantees.
2. The paper proposes a two-stage GNN framework. 1. A standard GNN is used to learn incremental community label estimation from the graph structure and initial features. 2. Using the output of the first stage as the initial features, another GNN is trained, simulating the local refinement step in classic methods, iteratively improving the prediction results by leveraging the label information of neighboring nodes.
3. The method was validated on both synthetic and real-world data, demonstrating significant advantages in theoretically challenging scenarios, and exhibiting strong robustness and generalization on mixed datasets.

**Weaknesses:**

1. The paper implies that all communities are of the same order, i.e., there are no communities significantly larger or smaller than others. In real networks, community sizes often follow a heavy-tailed distribution, resulting in extreme imbalance. Experimental results (e.g., the n^(4) results in Table 1) also show that the two-stage method’s performance fluctuates and degrades under extreme imbalance.

2. The authors manually construct a set of GNN parameters θ and show that this set can achieve the desired functionality. It is unclear whether training via stochastic gradient descent can find such a parameter set, whether it can avoid local optima to achieve (or approximately achieve) the theoretical solution. Furthermore, the GNNs used in experiments are much smaller than the theoretically constructed models, and the paper does not explain why the smaller models still exhibit theoretically guaranteed behavior (e.g., whether this relies on careful experimental design).

3. There is a lack of ablation studies on key factors such as the type of initialization Q0, GNN width/layers/sparsity, and whether training can reach the constructive solution.

4. There are some grammatical and punctuation errors, e.g., "an positive," and the formatting of mathematical symbols is inconsistent, e.g., x(0) vs. x (0).

**Questions:**

1. Could the assumptions mentioned above be relaxed? For example, is it possible to relax the assumption that $K$ is known and constant?
2. It would be better discuss the gap between existence proofs and practically trainable models. For example, explain why, even though the constructive GNN in theory is large, small-scale GNNs used in practice can still succeed.

---

### Official Review · Reviewer_zyur · 2025-11-02

**Soundness:** 2
**Presentation:** 1
**Contribution:** 2
**Rating:** 2
**Confidence:** 5

**Summary:**

This paper presents a rigorous theoretical and empirical analysis of a two-stage GNN framework for supervised community detection under the SBM. The authors establish, for the first time, that a suitably designed GNN can achieve information-theoretic optimality, matching the minimax misclassification rate of classical spectral and likelihood-based methods. The proposed method is shown to be robust to model misspecification and generalizes effectively to real-world networks, bridging a critical gap between deep learning practice and statistical theory in community detection.

**Strengths:**

(i) The paper's primary strength is its foundational theoretical contribution. It provides the first comprehensive statistical guarantees for GNN-based community detection, including error bounds for approximating orthogonal iteration (Theorem 1) and minimax-optimal misclassification rates for the final estimator (Theorems 2, 3).
(ii) The work successfully connects the power of GNNs with the well-established theoretical guarantees of classical algorithms. The two-stage framework is explicitly designed to emulate the theoretically optimal "spectral clustering + local refinement" pipeline, providing a principled deep learning alternative.
(iii) The empirical evaluation is thorough, spanning synthetic data from SBM, Degree-Corrected SBM (DCBM), and Latent Space Models (LSM), as well as multiple real-world networks. The experiments convincingly demonstrate the superiority of the two-stage approach, its robustness to model misspecification, and the benefit of mixed-model training.
(iv) The paper goes beyond in-sample performance and provides generalization bounds (Theorem 6), ensuring that the performance holds on out-of-sample graphs from the same generating process, a crucial aspect for practical applicability.

**Weaknesses:**

(i) The core theoretical results are derived for a relatively simple symmetric SBM (with only two parameters, $p$ and $q$. While the experiments show robustness to more complex models like DCBM, the theoretical analysis does not yet extend to these more realistic settings, which limits the direct applicability of the theoretical claims to real-world graphs with inherent degree heterogeneity.
(ii) The theoretical guarantees for the trained GNNs (Theorems 4 and 5) rely on the assumption that the training procedure can find parameters that achieve an empirical risk at least as good as the constructed GNNs. A formal analysis of the optimization landscape for this specific GNN architecture is left for future work, leaving a gap between the existence of a good solution and the ability of gradient-based methods to find it.
(iii) While the theoretical depth of the GNN is bounded polynomially in $log n$, the actual computational and memory cost of training and deploying such GNNs on large graphs is not discussed or compared against scalable classical methods, which could be a practical consideration for very large-scale applications.

**Questions:**

1. What is GRAPHICAL NEURAL NETWORKS in the title?
2.  The theoretical analysis is conducted for a simple two-parameter SBM. What are the primary technical hurdles in extending these optimality guarantees to more complex and realistic models like the Degree-Corrected SBM, and do you have a concrete pathway for doing so?
3. Theorems 4 and 5 assume the training process successfully minimizes the empirical risk. Can you comment on the optimization landscape of your proposed GNN architecture? Did you encounter any practical challenges in training, such as sensitivity to initialization or vanishing gradients, especially in the deeper networks required for the first stage?
4. The paper demonstrates strong empirical performance on real-world networks. However, these networks often exhibit features not captured by SBMs or DCBMs, such as overlapping communities or hierarchical structure. How do you envision adapting or extending your two-stage framework to handle such challenging scenarios?
5. In the mixed-model training experiments, performance on the LSM test set is notably lower than on SBM/DCBM. What is your interpretation of this performance gap? Does it suggest a fundamental limitation in the GNN's ability to capture latent space geometry, or is it a matter of training data diversity and model capacity?

---

### Official Review · Reviewer_AuRm · 2025-11-03

**Soundness:** 2
**Presentation:** 2
**Contribution:** 2
**Rating:** 6
**Confidence:** 3

**Summary:**

This paper provides a rigorous theoretical analysis of community detection with graph neural networks (GNNs) under the stochastic block model (SBM). The authors show that a carefully structured and trained two-stage GNN can attain information-theoretic optimality, matching the minimax rate achieved by classical spectral and likelihood-based estimators. Through approximation theory, the paper demonstrates how GNN layers emulate orthogonal iteration procedures, derives generalization bounds, and proposes a practical two-stage algorithm. Experiments on synthetic and real datasets confirm the theoretical findings, showing competitive or superior performance, particularly in low-signal regimes.

**Strengths:**

S1: Strong theoretical contribution—rigorously proves that GNNs can approximate spectral algorithms and achieve minimax misclassification rates under SBM.

S2: The approximation analysis is detailed and well-structured, explicitly characterizing error propagation, layer width, and depth requirements.

S3: Empirical validation is comprehensive, covering both controlled synthetic data and real-world networks, with consistent gains over spectral and likelihood baselines.

**Weaknesses:**

W1. Theoretical results are limited to standard SBMs with known, fixed K and assortative p>q. The framework does not yet extend to degree-corrected, overlapping, or dynamic settings that dominate practical applications.

W2. Key assumptions (e.g., $n(p-q) \gg \max\{\sqrt{np}, \sqrt{\log n}\}$) imply relatively strong signal regimes, excluding the hard-detection phase. This limits the generality of the optimality claim and should be discussed more explicitly.

W3. Some implementation details (e.g., selection of $\mathcal{F}(A)$, layer width/depth, and practical initialization of $Q_0$) are underspecified, making experimental reproducibility less transparent.

**Questions:**

Q1. How robust are the theoretical guarantees when extending to degree-corrected, overlapping, or dynamic community models, or when K is unknown?

Q2. Since theoretical results assume initialization close to the spectral subspace, how is this achieved in real datasets, and how sensitive is the method to initialization errors?

Q3. Have the authors empirically examined how approximation error evolves across GNN layers, complementing Figures 1–2’s architectural illustration?

---

### Meta-Review · Area_Chair_dp3q · 2025-12-25

**Summary:**

The reviewers raised numerous concerns about the paper, many of which the chair shares. None of them were answered by the authors.

**Reviewer Concerns:**

There was no rebuttal.

**Reviewer Scores:**

There was no rebuttal, no reason to change scores.

---

### Decision · Program_Chairs · 2026-01-26

Reject